# KLASS: KL-Guided Fast Inference in Masked Diffusion Models

Seo Hyun Kim[1][*]   Sunwoo Hong[1][*]   Hojung Jung[1]   Youngrok Park[1]   Se-Young Yun[1]

[1]KAIST AI

{shkimsally, sunwoo, yunseyoung}@kaist.ac.kr

## Abstract

Masked diffusion models have demonstrated competitive results on various tasks including language generation. However, due to its iterative refinement process, the inference is often bottlenecked by slow and static sampling speed. To overcome this problem, we introduce 'KL-Adaptive Stability Sampling' (KLASS), a fast yet effective sampling method that exploits token-level KL divergence to identify stable, high-confidence predictions. By unmasking multiple tokens in each iteration without any additional model training, our approach speeds up generation significantly while maintaining sample quality. On reasoning benchmarks, KLASS achieves up to $2.78\times$ wall-clock speedups while improving performance over standard greedy decoding, attaining state-of-the-art results among diffusion-based samplers. We further validate KLASS across diverse domains, including text, image, and molecular generation, showing its effectiveness as a broadly applicable sampler across different models. Our code is available at https://github.com/shkim0116/KLASS.

## 1 Introduction

Masked diffusion models [1, 28, 34, 38] have attracted growing attention for their ability to model joint distribution of sequences by iteratively refining samples from partially masked sequences to clean data, achieving competitive performance on complex language tasks [27], image generation [7], biological sequences [25, 34], and planning algorithms [50, 51].

Despite recent successes, these models are often restricted by slow and static sampling strategies such as Top-$k$ or stochastic sampling, where only a limited number of high-confidence tokens are unmasked at each step. As a result, the generation process can become inefficient and prone to local suboptimalities, thus constraining the practical applicability of masked diffusion approaches.

Several works investigate efficient samplers by caching the logits if no tokens are unmasked at the specific timestep [34] or design a specific scheduler to unmask one token at a time [56]. Another natural solution might be to incorporate an additional "planner" or auxiliary distribution to guide sampling [48, 55]. However, doing so typically incurs substantial computational overhead, increases inference latency, and can lead to difficulty aligning the planner's distribution with the base model's learned distribution. Instead, our goal is to develop a lightweight yet effective sampling method that remains within the model's own capabilities, yielding speedups in generation while simultaneously improving or maintaining overall accuracy.

To address these challenges, we propose 'KL-Adaptive Stability Sampling' (KLASS), an adaptive sampling strategy that leverages the diffusion model's own feedback to guide unmasking. Unlike previous approaches that rely on fixed schedules (i.e., a predetermined number of tokens unmasked at each timestep), our method adapts to the evolving confidence of the model during generation. We accelerate inference by identifying *stable* tokens as low-risk candidates for early unmasking. To

---

[*]Equal contribution

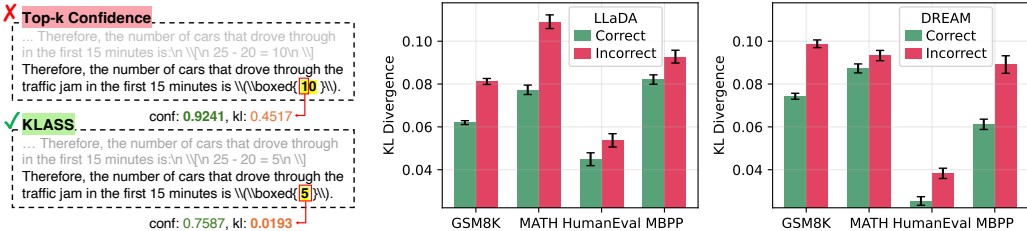

(a) Case study comparing Top-$k$ confidence and KLASS solutions.

(b) Average KL divergence of tokens at unmasking for correct and incorrect predictions on LLaDA and DREAM.

Figure 1: KL divergence as a strong indicator of solution correctness. (a) The Top-$k$ method selects an incorrect solution despite high confidence, whereas KLASS identifies the correct solution, which exhibits a significantly lower KL divergence. (b) KL divergence distributions for the LLaDA and DREAM models show that correct predictions consistently have lower KL divergence than incorrect ones across all datasets.

quantify stability, we track the token-level Kullback-Leibler (KL) divergence between conditional distributions at consecutive timesteps. Tokens are unmasked when their distributions remain similar (KL below a threshold) and are predicted with high confidence (probability exceeding a confidence threshold). This dynamic allocation of unmasking tokens results in significant acceleration of generation speed while maintaining sample quality by avoiding premature or suboptimal token unmasking without additional model training or extra memory burden.

We empirically validate our method on challenging reasoning benchmarks, including GSM8K, MATH, HumanEval, and MBPP. We show that applying KLASS with large-scale masked diffusion models not only halves the number of sampling steps compared to standard greedy or Top-$k$ decoding [19], but also achieves higher accuracy, achieving state-of-the-art results compared to other diffusion samplers. Figure 1a presents a comparison between solutions generated by Top-$k$ confidence and KLASS sampler. KLASS successfully identifies the correct token with lower KL, whereas Top-$k$ confidence tends to unmask incorrect tokens even with higher confidence. Furthermore, our experiment on plain text generation also proves the effectiveness of our method which results in reduced perplexity while maintaining entropy, thereby mitigating the inefficiencies inherent in conventional sampling. We further show that our sampler works in other modalities, including images and molecules.

Overall, our proposed sampler for masked diffusion models is both simple and practical, harnessing the latent potential of the base diffusion model itself, rather than relying on complex external planners. By strategically identifying stable tokens at each iteration, the algorithm accelerates generation and fosters more robust coverage of viable token candidates. We believe this work provides a practical and scalable way for large-scale masked diffusion models, particularly where reliable and efficient generation is essential, such as complex reasoning tasks.

We summarize our main contributions below:

- We propose KLASS, a training-free sampler that leverages the model's internal dynamics in terms of token level KL divergence and confidence without requiring external planners.

- We achieve up to $2.78\times$ faster sampling by more than halving the number of diffusion steps through parallel unmasking of stable tokens.

- We provide comprehensive empirical validation, showing improved quality on reasoning benchmarks across math and code generation, text generation, image synthesis, and molecular generation.

## 2 Related Works

**Discrete diffusion models** D3PM [1] investigate how forward and backward processes can be constructed in discrete state spaces which is analogous to the continuous diffusion models [18, 40]. [6] leverage continuous time Markov chain (CTMC) theory to formulate the forward-backward process of discrete diffusion models with providing negative ELBO in continuous time limit as an objective. Following the success of denoising score matching [41], Lou et al. [25], Meng et al. [26] suggest

discrete score matching loss by defining Stein score in discrete space. Ou et al. [28], Sahoo et al. [34], Shi et al. [38] further shows that simplified version of masked diffusion model can significantly boost the performance of diffusion models closing the performance gap with AR models in language domains. Recently, LLaDA [27] demonstrates scaling law of discrete diffusion models in language domain and further shows reasoning abilities.

**Discrete diffusion samplers**  Generating a text from language diffusion models involves iteratively refining a sequence from a noisy or masked state. Ancestral Sampling [25, 34] starts from a fully masked sequence and iteratively applies the learned reverse denoising process over a series of discrete timesteps to produce a clean sequence. SUBS parametrization [34] of the reverse step dictates how model predictions are used to unmask tokens, often by ensuring that already revealed tokens remain unchanged. To improve sample quality, ReMDM [43] adopts remasking strategies, where some newly predicted tokens are reset to a mask based on confidence or timestep.

**Accelerated Sampling of Discrete diffusion models**  The iterative nature of ancestral sampling can result in high latency due to the large number of sequential steps. Consequently, much research has focused on reducing the number of function evaluations (NFEs) in diffusion models. Deschenaux and Gulcehre [11], Hayakawa et al. [15] leverage distillation methods to train the model with reduced NFEs in analogous to fast sampling of continuous diffusion models [35, 41, 53]. Ren et al. [33] improve discrete diffusion solvers by considering second-order numerical solver in CTMC framework. Zheng et al. [56] propose a First-Hitting Sampler (FHS) to skip the unnecessary timesteps and unmask one token at a time. Most of the existing samplers of masked diffusion models, however, resort to additional training or rely on other models (i.e., planners) to choose unmasking tokens at each timestep [21, 24, 29]. This could help avoiding suboptimal token selection but with considerable computational overhead and may fail to be aligned with the model's intrinsic capability.

Recent training-free strategies for accelerating masked diffusion language models have emerged concurrently, with several works exploring heuristics based on model certainty to guide this process. Fast-dLLM [47] and Dimple [54] use confidence-aware decoding, SlowFast Sampling [45] alternates decoding stages based on certainty, convergence, and position principles, EB-Sampler [4] unmasks multiple tokens based on entropy bounds, and Prophet [23] uses the Top-2 confidence gap. While these concurrent approaches validate the utility of heuristics largely based on certainty, we empirically demonstrate that this signal alone is insufficient. To ensure tokens are not unmasked prematurely, we propose a novel method that utilizes KL divergence to identify stable tokens for parallel decoding.

## 3 Preliminaries

### 3.1 Masked diffusion models

In masked diffusion models, one requires an additional mask index $\mathbf{m}$ for each tokens and forward process is defined by following absorbing process [1]:

$$q(\mathbf{z}_t|\mathbf{x}) = \text{Cat}(\mathbf{z}_t; \alpha_t \mathbf{x} + (1 - \alpha_t)\mathbf{m}), \tag{1}$$

where $\alpha_t$ is predefined schedule, monotonically decreasing in $t$. Then one can analytically obtain posterior distribution as:

$$q(\mathbf{z}_s \mid \mathbf{z}_t, \mathbf{x}) = \begin{cases} \text{Cat}(\mathbf{z}_s; \mathbf{z}_t) & \text{if } \mathbf{z}_t \neq \mathbf{m}, \\ \text{Cat}\left(\mathbf{z}_s; \frac{(1-\alpha_s)\mathbf{m}+(\alpha_s-\alpha_t)\mathbf{x}}{1-\alpha_t}\right) & \text{if } \mathbf{z}_t = \mathbf{m}. \end{cases} \tag{2}$$

The goal of the masked diffusion model is to learn this reverse process by parameterizing the posterior (Eq. 2) by a neural network with $p_\theta(\mathbf{z}_s|\mathbf{z}_t) := q(\mathbf{z}_s|\mathbf{z}_t, \mu_\theta(\mathbf{z}_t, t))$.

In simplified masked diffusion models [28, 34, 38], learning objective can be simplified by parameterizing the models to focus on estimating only masked tokens while maintaining unmasked tokens throughout the generation.

Then the learning objective is to minimize Negative ELBO (NELBO) whose continuous form is the following:

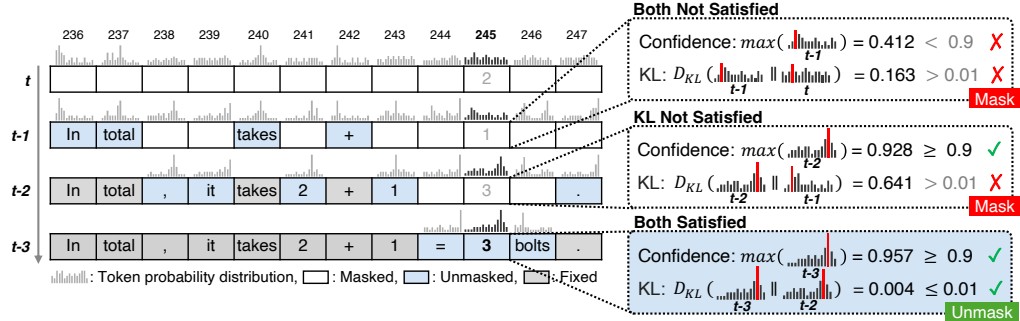

Figure 2: Illustration of parallel decoding with KLASS. Tokens are unmasked when they meet the two criteria: high predictive confidence and a stable probability distribution. Stability is measured by a low KL divergence between consecutive steps (illustrated with history length of 1 for simplicity). On the right it shows the sampling process for position 245: it remains masked due to low confidence or high KL score, and is unmasked when both conditions are satisfied.

$$\mathcal{L}_{\infty} = \mathbb{E}_{\mathbf{x} \sim q_0, \mathbf{z}_t \sim q_t(\mathbf{z}_t | \mathbf{x})} \int_0^1 \frac{\alpha'_t}{1 - \alpha_t} \left[ \delta_{\mathbf{x}, \mathbf{m}} \, \mathbf{x} \cdot \log \boldsymbol{\mu}_\theta(\mathbf{z}_t, t) \right]. \tag{3}$$

Here, $q_0$ denotes data distribution and $\alpha'_t$ is the derivative of noise schedule $\alpha_t$ in time. In this continuous time framework, [34] further proves that above objective is invariant of noise schedule $\alpha_t$.

The above can be generalized to sequence-level of token length $L$ modeling as follows.

$$\mathcal{L}_{\infty}^{(L)} = \int_0^1 \frac{\alpha'_t}{1 - \alpha_t} \mathbb{E}_{\mathbf{x} \sim q_0, \mathbf{z}_t \sim q_t(\mathbf{z}_t | \mathbf{x})} \left[ \sum_{l : \mathbf{z}_t^{(l)} = \mathbf{m}} \mathbf{x}^{(l)} \cdot \log \boldsymbol{\mu}_\theta^{(l)}(\mathbf{z}_t, t) \right] dt. \tag{4}$$

## 3.2 Inference via Ancestral Sampling

At inference, we discretize $t \in [0, 1]$ into times $\{t_T > \cdots > t_1 \approx 0\}$, initializing $\mathbf{x}_{t_T} = [\mathsf{mask}]^L$. We then sample backward:

$$\mathbf{x}_{t_{i-1}} \sim p_\theta(\mathbf{x}_{t_{i-1}} \mid \mathbf{x}_{t_i}), \quad i = T, \ldots, 1.$$

In simplified MDM, unmasked tokens remain fixed and masked tokens are drawn from the model's prediction. After $T$ steps, we obtain a complete sequence $\mathbf{x}_{t_0}$. We provide additional analysis of other sampling strategies in Appendix C.

## 4 Method

### 4.1 Defining Confidence Score and KL Score

KLASS aims to identify which tokens are stable enough to be unmasked at each step of the inference process, which we index by timesteps $t = T, \ldots, 1$. To guide this selection, we introduce two key metrics: a confidence score to measure the model's certainty on a given token and a KL score to measure the temporal consistency of its predictions.

**Definition 4.1.** *(Confidence score) Denoting $p_t^i$ as the categorical distribution predicted by the diffusion model at timestep $t$ for token position $i$, we define the confidence $\mathrm{conf}_t^i$ to be the largest value of the probability function among vocabulary space $V$ ($v \in V$):*

$$\mathrm{conf}_t^i = \max_v p_t^i(v). \tag{5}$$

Intuitively, a higher confidence score indicates the model is more certain about estimating the current token, which increases the chance that the model's estimate for that token is correct.

**Definition 4.2.** *(KL score)* *We define KL score $d_t^i$ of the token position $i$ at timestep $t$ as the Kullback-Leibler divergence between previous estimates and current estimates of the given token:*

$$d_t^i \;=\; D_{\mathrm{KL}}\big(p_t^i \,\|\, p_{t+1}^i\big), \tag{6}$$

*where we denote $p_t^i, p_{t+1}^i$ be the probability distribution of the model estimates of token index $i$ at time $t$ and at time $t+1$, respectively.*

KL score should be low only when the model's estimate is consistent throughout the reverse diffusion process, which implies the estimated token is more reliable.

To empirically demonstrate how KL score behaves in practical scenario, we first generate samples for a variety of math and programming reasoning benchmarks. As shown in Figure 1b, correct samples consistently exhibit significantly lower KL scores than incorrect ones, for all models and datasets. This observation motivates our use of KL scores as a guiding signal in the sampling algorithm of masked diffusion models, which we formally introduce in the next section.

### 4.2 KLASS: KL-Adaptive Stability Sampling

We introduce 'KL-Adaptive Stability Sampling' (KLASS), a novel sampling algorithm for masked diffusion models. As illustrated in Figure 2, KLASS leverages confidence score and KL score during the unmasking process of the masked diffusion models (Eq. 2), by selectively choosing unmasking tokens that have low KL score and high confidence score.

**Stable-token selection.** To effectively set the standard using both KL and confidence score, we propose stable-token selection in the following way: Given a history length $n$, a KL threshold $\epsilon_{\mathrm{KL}}$, and a confidence threshold $\tau$, we select the set of *stable* tokens at step $t$ as,

$$S_t = \Big\{ i \;\Big|\; \underbrace{\forall k \in \{1, \ldots, n\} \quad D_{\mathrm{KL}}\big(p_{t+k-1}^i \,\|\, p_{t+k}^i\big) < \epsilon_{\mathrm{KL}}}_{\text{all recent KL's below threshold}} \wedge \underbrace{\mathrm{conf}_t^i > \tau}_{\text{high confidence}} \Big\}. \tag{7}$$

**Unmasking rule.** KLASS adaptively chooses which tokens to unmask at given timestep with above defined stable index (Eq. 7). At each diffusion step $t$, we apply

$$x_t^i = \begin{cases} \text{unmask token at position } i, & i \in S_t, \\ \text{otherwise, unmask the Top-}u \text{ positions by } \mathrm{conf}_t^i, & S_t = \emptyset, \end{cases} \tag{8}$$

where $u$ is a fixed fallback unmasking count. We provide a pseudocode of our algorithm with further analysis in Appendix B.

## 5 Theoretical Rationale

We provide a theoretical perspective on why using KL divergence can improve sample quality. We show that, for a well-trained model, a token that is predicted as *incorrect* at the current step cannot remain uniformly stable as the context is progressively resolved.

**Definition 5.1.** *For each context $c$ (instantiation of variables outside $X_i$), let $\mathcal{C}(c)$ be the nonempty set of task–correct conditionals. Let $\mathcal{C} := \{\mu : \mu(\cdot \mid c) \in \mathcal{C}(c) \;\forall c\}$. We say $p_\theta$ is a conditional $\delta$–approximation to the task if*

$$\inf_{\pi \in \mathcal{C}} \sup_c \mathrm{TV}\big(p_\theta(\cdot \mid c), \pi(\cdot \mid c)\big) \;\leq\; \delta.$$

**Definition 5.2.** *Fix $i$. Let $x_i^\star$ be optimal under $\pi(\cdot \mid c^\star)$ at near-optimal context $c^\star$. Let $x_i^\dagger \neq x_i^\star$ be suboptimal. Assume a true margin $\gamma > 0$ at $c^\star$: $\pi(x_i^\star \mid c^\star) \geq \pi(x_i^\dagger \mid c^\star) + \gamma$. Assume the model currently prefers $x_i^\dagger$ at $c_M$ by margin $\beta \geq 0$: $p_\theta(x_i^\dagger \mid c_M) \geq p_\theta(x_i^\star \mid c_M) + \beta$.*

**Proposition 5.3.** *Suppose $p_\theta$ is a conditional $\delta$-approximation of $\pi$. For any context path $c_M \to c_{M-1} \to \cdots \to c_0$ (changing only variables outside $X_i$) ending at $c_0 = c^\star$, let $P_t := p_\theta(\cdot \mid c_t)$ and $\Delta := \frac{1}{2}(\beta + \gamma - 2\delta)_+$. Then*

$$\mathrm{TV}(P_M, P_0) \;\geq\; \Delta, \qquad \frac{1}{M} \sum_{t=0}^{M-1} \mathrm{KL}\big(P_t \,\|\, P_{t+1}\big) \;\geq\; \frac{2\,\Delta^2}{M^2}.$$

*Proof.* The proof is in Appendix A. $\qquad\qquad\square$

Table 1: Performance and sampling steps on reasoning benchmarks for different diffusion samplers.

| Method | Parallel | MATH | | GSM8K | | HumanEval | | MBPP | |
|--------|----------|------|------|-------|------|-----------|------|------|------|
| | | Acc↑ | Steps↓ | Acc↑ | Steps↓ | Acc↑ | Steps↓ | Acc↑ | Steps↓ |
| **LLaDA** | | | | | | | | | |
| Top-1 | ✗ | 31.4 | 256 | 75.13 | 256 | 39.63 | 256 | 46.69 | 256 |
| Random | ✗ | 26.2 | 256 | 67.10 | 256 | 20.21 | 256 | 29.18 | 256 |
| Top-2 | ✓ | 29.6 | 128 | 72.40 | 128 | 33.54 | 128 | 37.74 | 128 |
| Confidence | ✓ | 31.6 | 96.46 | 75.21 | 74.35 | 37.80 | 54.41 | 47.08 | 85.20 |
| KL divergence | ✓ | 32.6 | 172.21 | 74.52 | 155.88 | 40.24 | 111.93 | 45.53 | 150.47 |
| KLASS (ours) | ✓ | **33.8** | 128.62 | **76.50** | 98.57 | **40.85** | 91.98 | **47.86** | 119.59 |
| **Dream** | | | | | | | | | |
| Top-1 | ✗ | 37.97 | 256 | **79.55** | 256 | 58.53 | 256 | 63.81 | 256 |
| Random | ✗ | 18.73 | 256 | 37.35 | 256 | 18.09 | 256 | 28.14 | 256 |
| Top-2 | ✓ | 33.60 | 128 | 71.69 | 128 | 42.88 | 128 | 47.08 | 128 |
| Confidence | ✓ | 41.80 | 95.10 | 73.67 | 74.81 | 50.00 | 52.47 | 57.59 | 72.49 |
| KL divergence | ✓ | 41.27 | 162.49 | 76.70 | 150.02 | **59.35** | 73.94 | 62.65 | 108.15 |
| KLASS (ours) | ✓ | **43.20** | 149.72 | 79.43 | 155.67 | **59.35** | 74.88 | **64.59** | 111.24 |

**Remarks.** A token that is wrong at $c_0$ but correct at $c^\star$ must be *dynamically unstable* somewhere along the path: its average per-step KL is bounded away from 0. In essence, incorrect tokens cannot remain dynamically stable. Accordingly, KLASS delays unmasking until tokens exhibit dynamic stability thereby improving generation quality.

# 6 Experiments

To show effectiveness of our proposed sampler, we conduct experiments on multiple benchmarks including reasoning benchmarks with large scale models in Section 6.1, text generation in Section 6.2, along with other modalities including images in Section 6.3 and molecules in Section 6.4. We also present ablation studies in Section 6.5 and analyze computational overhead in Section 6.6.

## 6.1 Reasoning tasks

**Experimental setup** We evaluate on four reasoning benchmarks: GSM8K [10] and MATH500 [16] for math, and HumanEval [9] and MBPP-sanitized [2] for code synthesis. We use two instruction-tuned models, LLaDA 8B Instruct [27] and Dream 7B Instruct [52]. For both models we set the generation length to 256 tokens, with LLaDA using a block size of 64. The generation temperature is set to 0 for LLaDA and 0.2 for Dream. We report both the number of sampling steps and the pass@1 accuracy. The maximum inference timestep is set to 256. In KLASS, we compute per-token KL divergence over a history length of $n = 2$, and apply KL thresholds ranging from 0.001 to 0.01 and confidence thresholds from 0.5 to 0.9. Full configuration details and a lightweight guideline for hyperparameter selection are provided in Appendix D.1.2.

**Baselines** We compare KLASS against baselines across two categories. The first is sequential unmasking (single-token), which includes: (i) Top-1 sampling, selecting the highest-confidence token at each step [7]; and (ii) random sampling [1]. The second category is parallel unmasking, which accelerates generation by revealing multiple tokens per step: (iii) Top-2 sampling, decoding the two highest-confidence tokens per step to halve the total number of steps; (iv) confidence-threshold sampling, unmasking all tokens with a predicted probability over 0.9; and (v) KL-threshold sampling, unmasking all tokens with a KL divergence under 0.001, using a history length $n = 2$ as in KLASS.

**Results** As shown in Table 1, KLASS consistently improves accuracy across most tasks compared to the standard greedy decoding (Top-1) baseline. It demonstrates robust generalization for both LLaDA and Dream models across math and code synthesis benchmarks. Beyond accuracy, KLASS

Table 2: Generative perplexity, MAUVE, and entropy on unconditional text generation sampled with 512 steps.

| Method | MAUVE↑ | LLaMA2↓ | LLaMA3↓ | GPT-2↓ | Entropy↑ |
|---|---|---|---|---|---|
| *Data | 1.0 | 7.0 | 9.4 | 14.8 | 5.44 |
| AR | 0.855 | 10.97 | 15.12 | 12.07 | 5.21 |
| SEDD | 0.037 | 53.09 | 109.60 | 105.40 | **5.62** |
| D3PM | 0.022 | 41.82 | 72.85 | 76.70 | 5.40 |
| MDLM | 0.115 | 30.88 | 54.15 | 51.78 | 5.46 |
| KLASS (Ours) | **0.179** | **26.94** | **49.19** | **45.50** | 5.43 |

Table 3: Generative FID and IS on MMaDA with different step sizes.

| Method | Steps | FID↓ | IS↑ |
|---|---|---|---|
| Confidence | 16 | 34.48 | 75.72 |
| KLASS (ours) | 16 | **30.48** | **93.07** |
| Confidence | 32 | 36.45 | 72.40 |
| KLASS (ours) | 32 | **32.00** | **89.17** |

Table 4: Molecular generation results on the QM9 dataset conditioned on different molecular properties.

| Method | Property | Reward↑ | NFEs↓ |
|---|---|---|---|
| MDLM | QED | 0.526 | 32.0 |
| KLASS (ours) | QED | **0.546** | 18.8 |
| MDLM | Ring count | 4.123 | 32.0 |
| KLASS (ours) | Ring count | **4.258** | 24.4 |

is also highly efficient. It reduces sampling steps by 40–70% relative to the full 256-step schedule, yielding wall-clock speedups of up to $2.78\times$ (Appendix D.1.3). Unlike other acceleration strategies such as halving steps with a confidence-based Top-2 method, which degrades accuracy, KLASS improves accuracy with fewer steps overall. KLASS achieves a superior balance between speed and accuracy compared to methods that rely on a single threshold for either confidence or KL score alone. This proves that the effectiveness of KLASS comes from its novel approach of combining token confidence with KL-divergence trajectories.

## 6.2 Text generation

**Experimental setup** We evaluate KLASS on Masked Diffusion Language Model (MDLM) [34] pre-trained on the OpenWebText corpus [13]. As baselines, we include (i) the original autoregressive sampler, (ii) SEDD [25], and (iii) two variants of MDLM: one parameterized with SUBS (the standard 512-step sampler) and one parameterized with D3PM [3] (the absorbing variant). For all diffusion-based methods, we generate 1,000 sequences of length 1,024 tokens under a fixed 512-step schedule, with nucleus Top-$p$ filtering at $p = 0.9$, history length $n = 2$, KL threshold $\epsilon_{\text{KL}} = 1e-4$, and confidence threshold $\tau = 0.57$.

**Evaluation** We report generative perplexity by exponentiating the average token-level negative log-likelihood under three oracle models: LLaMA2 (7B) [42], LLaMA3 (8B) [14], and GPT-2 [30]. We measure Shannon entropy of the predicted token distributions and compute MAUVE by comparing our 1,000 generated samples to 1,000 held-out segments from the OpenWebText. Baseline (*Data) results are given from the corresponding literatures [43, 48].

**Results** Table 2 shows that KLASS substantially improves generative quality over existing discrete diffusion samplers. Our method higher MAUVE and lower perplexity across all oracle models while maintaining comparable entropy. These results highlight that stability-aware multi-token unmasking guided by KLASS leads to more coherent and fluent text generation, all without any additional model training. We provide experimental details in Appendix D.2.

## 6.3 Image generation

**Experimental setup** We evaluate KLASS on the MMaDA (Multimodal Large Diffusion Language Models) [49], a multimodal diffusion foundation model. We compare two samplers: (i) the standard confidence-based sampler used by MMaDA, and (ii) our proposed KLASS. For each method, we

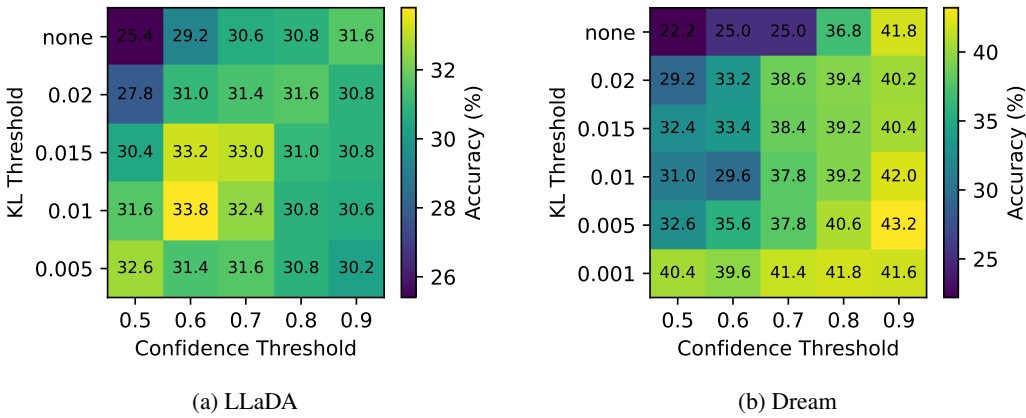

(a) LLaDA                                    (b) Dream

Figure 3: KL Effect Across Confidence Levels on MATH.

generate 10,000 images conditioned on labels drawn uniformly from the 1,000 ImageNet classes, using 16 and 32 step decoding schedules. KLASS is configured with history length $n = 1$, KL divergence threshold $\epsilon_{KL} = 0.3$, and confidence threshold $\tau = 0.1$.

**Evaluation**   We assess sample fidelity using two widely adopted metrics. First, we compute Fréchet Inception Distance (FID) [17] between our 10,000 generated samples and the ImageNet validation set, using the official Inception v3 implementation. Second, we measure Inception Score (IS) [36] on the same samples with the standard protocol.

**Results**   Table 3 shows that KLASS improves image quality on MMaDA over the standard confidence-based sampler. Across both decoding schedules, KLASS yielding lower FID and higher IS. The trend holds under the same decoding schedules and fairness controls, indicating that KLASS improves fidelity and class-consistency without modifying the backbone or adding auxiliary guidance. We provide experimental details in Appendix D.3.

## 6.4   Molecular generation

**Experimental setup**   We use QM9 [31], which contains molecules with up to nine heavy atoms, represented in SMILES [46]. For models we follow the training recipe of [37] to train seperate models conditioned on drug-likeness (QED) [5] and number of rings using classifier-free training of masked diffusion models.

**Evaluation**   We test KLASS on conditional generation of small molecules using CFG guidance. Specifically, we aim to generate molecules with higher score of QED or maximizing ring counts while fixing the CFG strength for fair comparison. We generate 1,024 samples for each task and provide average value of number of function evaluation (NFEs). Further details of the experimental setups are provided in Appendix D.4.

**Results**   The result shows that KLASS effectively reduces the total sampling steps while maintaining target reward in the conditional generation scenario for both target reward (QED and Ring count). We provide further experimental results in this setup in Appendix D.4.

## 6.5   Ablation Studies

**Effect of confidence and KL score thresholds**   Our evaluation of different confidence and KL thresholds on the MATH dataset reveals that combining both is essential for optimal performance. As shown in Figure 3, applying the KL threshold consistently enhances accuracy across all confidence levels compared to relying on a confidence threshold alone ('none' row). This synergistic relationship is further substantiated by Table 1, which demonstrates that using a single criterion leads to a notable reduction in accuracy.

Table 5: Comparison of single-token and parallel unmasking strategies under KLASS criteria.

| Unmasking | MATH | | GSM8K | |
|---|---|---|---|---|
| | Acc $\uparrow$ | Steps $\downarrow$ | Acc $\uparrow$ | Steps $\downarrow$ |
| Single (conf) | 31.2 | 256 | 72.86 | 256 |
| Single (KL) | 29.0 | 256 | 73.46 | 256 |
| Parallel | **33.8** | 128.6 | **76.50** | 98.57 |

Table 6: Memory and computational overhead of KL divergence per decoding step.

| Model | Memory (MB) | | Time (s) | |
|---|---|---|---|---|
| | Overhead | Total | Overhead | Total |
| **LLaDA** | 247 | 18,702 | 0.000255 | 0.1218 |
| **Dream** | 296 | 18,875 | 0.000177 | 0.1275 |

While the optimal hyperparameter settings vary significantly between models, each model's performance remains stable and robust around its unique optimal point. For example, LLaDA performs best with a lower confidence threshold, whereas Dream requires a higher one to achieve maximum accuracy. In both cases, however, accuracy does not degrade sharply near these values, indicating low sensitivity to minor hyperparameter adjustments. A more detailed sensitivity analysis, featuring additional tasks and a finer-grained grid of thresholds, is provided in Appendix D.5.1.

**Effect of unmasking multiple tokens** We evaluate whether unmasking multiple tokens per step improves LLaDA's performance. Using KLASS, which selects tokens based on fixed thresholds, we compare parallel multi-token unmasking to two sequential variants. These variants unmask only a single token from the same stable pool satisfying the KLASS criteria: 'Single (conf)' unmasks the one with the highest confidence and 'Single (KL)' unmasks the one with the lowest KL score.

As shown in Table 5, parallel sampling of KLASS boosts both accuracy and efficiency. On MATH, it improves accuracy by up to 4.8 points while cutting sampling steps by nearly 50%. Similar trends hold on GSM8K. These results suggest that LLaDA benefits from unmasking multiple stable tokens in parallel, leading to faster and even more accurate reasoning.

## 6.6 Analysis on Computational Overhead

The overhead of KL computation is negligible, as it is a lightweight post-processing step on existing logits that requires no additional forward pass. For the set of masked tokens $I_m = \{i \mid z_t^i = m\}$, we compute the KL score $d_t^i = D_{KL}(p_t^i \| p_{t+1}^i)$ and cache the prior distribution. This yields a combined computational and memory overhead of $O(|I_m| \cdot |V|)$, a linear cost that is negligible compared to the expensive matrix multiplications and multi-gigabyte footprint of the main diffusion step. Table 6 empirically supports this conclusion. We measure the overhead for LLaDA and Dream, with vocab sizes of 126,464 and 152,064, respectively, using a generation length of 256. The results show memory overheads below 1.57% of total memory and latency overheads below 0.21% per decoding step, confirming that KL computation adds only minimal cost.

## 7 Conclusion

We proposed KL-Adaptive Stability Sampling (KLASS), an efficient and adaptive sampling method for masked diffusion models that leverages token-level KL divergence and model confidence to guide the unmasking process. KLASS substantially reduces the number of sampling steps while maintaining or improving accuracy, achieving state-of-the-art performance on math and code reasoning benchmarks. Our approach is simple, requires no additional training, and generalizes well across multiple modalities, making it a practical solution for faster and more reliable generation in masked diffusion models.

For future work, one could extend this approach to discrete diffusion models with alternative noise schedules, such as the uniform or marginal prior [1]. Another direction is to evaluate the proposed sampler with larger models as they become available. We also discuss the broader impact and limitations of our work in Appendix G.

## Acknowledgments

This work was supported by Institute of Information & Communications Technology Planning & Evaluation (IITP) grant funded by the Korea government (MSIT) (RS-2024-00457882, AI Research Hub Project, 10%) and (RS-2022-II220311, Development of Goal-Oriented Reinforcement Learning Techniques for Contact-Rich Robotic Manipulation of Everyday Objects, 90%).

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

# Table of contents

# A    Theoretical Proofs

In this section, we provide the proof for Section 5.

**Definition A.1.** *For each context c (instantiation of variables outside $X_i$), let $\mathcal{C}(c)$ be the nonempty set of task–correct conditionals. Let $\mathcal{C} := \{\mu : \ \mu(\cdot \mid c) \in \mathcal{C}(c) \ \forall c\}$. We say $p_\theta$ is a conditional $\delta$–approximation to the task if*

$$\inf_{\pi \in \mathcal{C}} \ \sup_c \ \mathrm{TV}\big(p_\theta(\cdot \mid c), \pi(\cdot \mid c)\big) \ \leq \ \delta.$$

**Definition A.2.** *Fix i. Let $x_i^\star$ be optimal under $\pi(\cdot \mid c^\star)$ at near-optimal context $c^\star$. Let $x_i^\dagger \neq x_i^\star$ be suboptimal. Assume a true margin $\gamma > 0$ at $c^\star$: $\pi(x_i^\star \mid c^\star) \geq \pi(x_i^\dagger \mid c^\star) + \gamma$. Assume the model currently prefers $x_i^\dagger$ at $c_M$ by margin $\beta \geq 0$: $p_\theta(x_i^\dagger \mid c_M) \geq p_\theta(x_i^\star \mid c_M) + \beta$.*

**Proposition A.3.** *Suppose $p_\theta$ is a conditional $\delta$-approximation of $\pi$. For any context path $c_M \to c_{M-1} \to \cdots \to c_0$ (changing only variables outside $X_i$) ending at $c_0 = c^\star$, let $P_t := p_\theta(\cdot \mid c_t)$ and $\Delta := \frac{1}{2}(\beta + \gamma - 2\delta)_+$. Then*

$$\mathrm{TV}(P_M, P_0) \ \geq \ \Delta, \qquad \frac{1}{M} \sum_{t=0}^{M-1} \mathrm{KL}\big(P_t \parallel P_{t+1}\big) \ \geq \ \frac{2\,\Delta^2}{M^2}.$$

*Proof.* Let $f = \mathbf{1}\{x_i = x_i^\dagger\} - \mathbf{1}\{x_i = x_i^\star\}$ so $\|f\|_\infty \leq 1$. Then

$$2\,\mathrm{TV}(P_M, P_0) \ \geq \ \big| \mathbb{E}_{P_0}[f] - \mathbb{E}_{P_M}[f] \big| = \big| (p_\theta^\dagger(c_M) - p_\theta^\star(c_M)) - (p_\theta^\dagger(c^\star) - p_\theta^\star(c^\star)) \big|.$$

By the margin assumptions (Definition A.2) and the conditional $\delta$–approximation (Definition A.1), $p_\theta^\dagger(c_M) - p_\theta^\star(c_M) \geq \beta$ and $p_\theta^\star(c^\star) - p_\theta^\dagger(c^\star) \geq \gamma - 2\delta$. Hence $2\,\mathrm{TV}(P_M, P_0) \geq \beta + \gamma - 2\delta$, which implies $\mathrm{TV}(P_M, P_0) \geq \Delta$.

By the triangle inequality in total variation,

$$\sum_{t=0}^{M-1} \mathrm{TV}(P_{t+1}, P_t) \ \geq \ \mathrm{TV}(P_M, P_0) \ \geq \ \Delta.$$

Let $T_t := \mathrm{TV}(P_{t+1}, P_t)$. Pinsker's inequality gives $\mathrm{KL}(P_t \| P_{t+1}) \ \geq \ 2\,T_t^2$ for each $t$ (since $TV(P_{t+1}, P_t) = TV(P_{t+1}, P_t)$). Averaging and applying Cauchy–Schwarz,

$$\frac{1}{M} \sum_{t=0}^{M-1} \mathrm{KL}(P_t \| P_{t+1}) \ \geq \ \frac{2}{M} \sum_{t=0}^{M-1} T_t^2 \ \geq \ \frac{2}{M} \cdot \frac{\big(\sum_{t=0}^{M-1} T_t\big)^2}{M} \ \geq \ \frac{2\,\Delta^2}{M^2}.$$

$\square$

# B    Pseudo Code

We provide the pseudo code for our KLASS algorithm, implementing the unmasking rule described in Section 4.2.

# C    Further Prior Works

In this section, we review several approaches for sampling from discrete diffusion models described in prior work.

**Ancestral sampling**    Generation proceeds by discretizing the reverse diffusion time-interval $[0, 1]$ into

$$0 = t_0 < t_1 < \cdots < t_T = 1.$$

To sample a sequence of length $L$, one initializes

$$z_{1:L}^{(T)} \ = \ [\texttt{mask}]^L,$$

**Algorithm 1: KL-Adaptive Stability Sampling (KLASS)**

---

**Input:** model $M$, total steps $T$, sequence length $L$, confidence threshold $\tau$, KL threshold $\epsilon$, history window $H$, fallback count $u$

**Output:** Generated sequence $\mathbf{x}$

1   **Initialize** $\mathbf{x} \leftarrow [\text{MASK}_{1:L}]$, $\mathbf{P}_{\text{prev}} \leftarrow 0$, $\mathbf{KLbuf} \leftarrow 0$

2   **for** $t \leftarrow 1$ **to** $T$ **do**

3     $\boldsymbol{\ell} \leftarrow M(\mathbf{x}).\text{logits}$

4     $\mathbf{P} \leftarrow \text{softmax}(\boldsymbol{\ell})$

5     $\mathbf{c} \leftarrow \max(\mathbf{P})$  // Get per-token confidence scores

6     $\boldsymbol{\delta} \leftarrow D_{\text{KL}}(\mathbf{P} \,\|\, \mathbf{P}_{\text{prev}})$  // Get per-token KL scores

     // Update KL history buffer

7     $\mathbf{KLbuf} \leftarrow \text{roll}(\mathbf{KLbuf}, \text{shift} = -1)$

8     $\mathbf{KLbuf}[:H] \leftarrow \boldsymbol{\delta}$

9     $\mathbf{P}_{\text{prev}} \leftarrow \mathbf{P}$

     // Identify stable tokens

10    $\mathbf{stable\_kl} \leftarrow \forall(\mathbf{KLbuf} < \epsilon)$  // Check all history

11    $\mathbf{high\_conf} \leftarrow (\mathbf{c} > \tau)$

12    $\mathbf{is\_masked} \leftarrow \text{isMask}(\mathbf{x})$

13    $\mathbf{ready} \leftarrow \mathbf{stable\_kl} \wedge \mathbf{high\_conf} \wedge \mathbf{is\_masked}$

14    **if** any($\mathbf{ready}$) **then**

15      unmask_at_indices($\mathbf{x}, \mathbf{P}, \mathbf{ready}$)

16    **else**

      // Fallback: unmask top-$u$ tokens

17      $\mathbf{scores} \leftarrow \mathbf{c} \cdot \mathbf{is\_masked}$  // Zero out unmasked tokens

18      $U \leftarrow \text{topk\_indices}(\mathbf{scores}, u)$

19      unmask_at_indices($\mathbf{x}, \mathbf{P}, U$)

20   **Return** $\mathbf{x}$

---

and for $i = T, T-1, \dots, 1$ draws each coordinate independently:

$$z_\ell^{(i-1)} \;\sim\; \begin{cases} \delta\big(z_\ell^{(i)}\big), & z_\ell^{(i)} \neq \texttt{mask}, \\ p_\theta\big(z_\ell \mid z_{1:L}^{(i)}\big), & z_\ell^{(i)} = \texttt{mask}, \end{cases} \quad \ell = 1, \dots, L.$$

Because unmasked tokens remain unchanged, if at step $i$ no new tokens are decoded then $z_{1:L}^{(i-1)} = z_{1:L}^{(i)}$, and—when the denoiser $\mu_\theta$ is time-invariant—its output at $t_i$ can be reused at $t_{i-1}$, skipping that network evaluation [28, 34]. Models whose $\mu_\theta$ depends on $t$ (e.g. SEDD [25]) must recompute at every $t_i$ and cannot exploit this caching [25, 34, 38].

**Exact simulation**   Exact simulation of the reverse CTMC in absorbing masked diffusion is achieved via uniformization: one bounds the time-varying generator $Q(t)$ by $\lambda$, samples

$$M \sim \text{Poisson}(\lambda T), \qquad \{\tau_i\}_{i=1}^{M} \overset{\text{iid}}{\sim} \text{Unif}(0, T),$$

and at each $\tau_i$ transitions from state $x$ to $y$ with probability $Q_{x,y}(\tau_i)/\lambda$, preserving the exact law of the reversed path [8]. While unbiased, as the chain nears absorption the number of proposals—and hence cost—can grow large.

Alternatively, the first-hitting sampler of Zheng et al. [56] draws each unmasking time without discretization error: when $n$ tokens remain masked, one samples

$$\tau_{n-1} = \alpha^{-1}\Big(1 - u_n^{1/n}\,[1 - \alpha(\tau_n)]\Big), \quad u_n \sim Unif(0, 1),$$

then un-masks exactly one token (chosen uniformly among the $n$) according to the model's conditional distribution. This procedure is unbiased but incurs $O(L)$ sequential events, making runtime scale with sequence length [56].

$\tau$**-leaping**   Tau-leaping discretizes the reverse CTMC into fixed intervals of length $\tau$, holding all jump rates constant and applying transitions in parallel. Let $\widehat{R}_t^\theta(x, x')$ be the learned rate from $x$ to $x'$ at time $t$. Over $[t - \tau, t]$ one draws

$$K_{x \to x'} \;\sim\; \text{Poisson}\big(\tau\,\widehat{R}_t^\theta(x, x')\big),$$

and updates

$$x_{t-\tau} = x_t + \sum_{x' \neq x_t} K_{x_t \to x'} \left( e_{x'} - e_{x_t} \right).$$

Under mild regularity, the global weak error scales as $O(\tau)$, recovering Gillespie's exact algorithm when $\tau \to 0$ [12]. Ren et al. derive a KL divergence bound

$$D_{\mathrm{KL}}\left(\mathcal{L}_{\tau\text{-leap}} \parallel p_{\mathrm{data}}\right) \leq O(\tau\,T) + O(M\,T) + O\!\left(e^{-cT/\log D}\right),$$

showing modest $\tau$ suffices even in high dimensions [32]. In practice, coordinates with multiple jumps are rejected to enforce categorical integrity (a negligible event under suitable rates), trading $O(1)$ network evaluations per leap for an $O(\tau)$ discretization bias.

**High-order samplers**   To improve on first-order $\tau$-leaping, multi-stage integrators achieve higher local accuracy. Ren et al. [33] introduce two-stage schemes with second-order convergence in KL. The $\theta$-RK-2 method computes an intermediate state

$$y^* = y_t + \sum_{\nu \in D} \nu\,\mathrm{Poisson}\!\left(\mu_t(\nu)\,\theta\,\Delta t\right),$$

then updates

$$y_{t-\Delta t} = y_t + \sum_{\nu \in D} \nu\,\mathrm{Poisson}\!\left(\left(1 - \tfrac{1}{2\theta}\right)\mu_t(\nu) + \tfrac{1}{2\theta}\,\mu_t^*(\nu)\right)\Delta t,$$

where $\mu_t^*$ is the intensity at $y^*$. The $\theta$-trapezoidal variant replaces the second stage with

$$y_{t-\Delta t} = y^* + \sum_{\nu \in D} \nu\,\mathrm{Poisson}\!\left((\alpha_1\,\mu_t^* - \alpha_2\,\mu_t)(\nu)\,(1 - \theta)\,\Delta t\right),$$

with $\alpha_1 = \frac{1}{2\theta(1-\theta)}$, $\alpha_2 = \frac{(1-\theta)^2 + \theta^2}{2\theta(1-\theta)}$. These reduce the discretization error to $O(\Delta t^2 T)$, enabling 3–5× fewer evaluations for comparable fidelity [33], and draw on high-order continuous schemes [20].

# D   Experiment details and additional results

## D.1   Reasoning tasks

### D.1.1   Experiment details

We conduct our experiments using LLaDA 8B Instruct [27] and Dream 7B Instruct [52], which are masked diffusion models capable of advanced reasoning. Across all settings, we fix the generation length to 256 tokens. For LLaDA 8B Instruct, which supports semi-autoregressive sampling, we set the block size to 64 for all experiments. Sampling temperature is set to 0 for LLaDA 8B Instruct, and to 0.2 for Dream 7B Instruct. Additional experiments using Dream 7B Instruct with a temperature of 0 are reported in Appendix D.5.3. All sampling experiments are conducted on a single NVIDIA RTX A5000 GPU.

### D.1.2   Hyperparameter Selection Guideline

For hyperparameter selection, we adopt a lightweight three-step search procedure using a small validation set of around 100 examples.

1. **Initial KL Threshold Estimation:** We obtain an initial estimate of the KL threshold by inspecting the distribution of KL values during decoding.
2. **Confidence Threshold Search:** With the KL threshold fixed, we sweep confidence values (0.9 down to 0.6) to identify the best trade-off between accuracy and decoding speed.
3. **KL Threshold Refinement:** With the confidence threshold fixed, we refine the KL threshold through a finer-grained search around the initial estimate.

This procedure is efficient, requiring only a small number of validation samples and negligible computation relative to training. The final confidence and KL threshold configurations for each dataset and model are summarized in Table 7.

Table 7: Experiment threshold configurations for KLASS

| Model | MATH | | GSM8K | | HumanEval | | MBPP | |
|-------|------|------|-------|------|-----------|------|------|------|
| | Conf | KL | Conf | KL | Conf | KL | Conf | KL |
| LLaDA | 0.6 | 0.010 | 0.6 | 0.015 | 0.9 | 0.010 | 0.7 | 0.010 |
| Dream | 0.9 | 0.005 | 0.9 | 0.001 | 0.8 | 0.001 | 0.9 | 0.001 |

Table 8: Wall-clock time per sample for Top-1 and KLASS decoding.

| Model | Dataset | Accuracy | | Time (s) | | Speedup |
|-------|---------|----------|-------|----------|-------|---------|
| | | Top-1 | KLASS | Top-1 | KLASS | |
| **LLaDA** | GSM8K | 75.13 | 76.50 | 37.04 | 15.86 | 2.34× |
| | MATH | 31.40 | 33.80 | 38.40 | 21.41 | 1.79× |
| | HumanEval | 39.63 | 40.85 | 39.54 | 16.04 | 2.47× |
| | MBPP | 46.69 | 47.86 | 39.12 | 20.68 | 1.89× |
| **Dream** | GSM8K | 79.75 | 80.44 | 29.66 | 22.26 | 1.33× |
| | MATH | 38.00 | 43.20 | 30.76 | 23.31 | 1.32× |
| | HumanEval | 58.53 | 59.76 | 32.01 | 11.52 | 2.78× |
| | MBPP | 63.81 | 64.59 | 31.89 | 17.65 | 1.81× |

### D.1.3 Wall-clock time comparison

We compare the actual wall-clock time of the KLASS and Top-k samplers on the MATH dataset in Table 8. Compared to the Top-k sampler with 256 steps (i.e., unmasking one token per step), KLASS reduces generation time by 47.4% for LLaDA and by 16.1% for Dream).

When comparing KLASS with the Top-k sampler using a similar number of steps (128 for LLaDA and 149 for Dream), the Top-k method achieves slightly lower generation times. However, this speed gain comes at the cost of reduced accuracy. KLASS not only maintains a comparable runtime but also improves accuracy, demonstrating its efficiency and effectiveness.

### D.1.4 Statistical significance of Dream 7B Instruct results

In Table 9, we report the mean and standard deviation over three runs for all methods using Dream 7B Instruct with a temperature of 0.2. Since the experiments with LLaDA 8B Instruct were conducted with a temperature of 0, the results are deterministic, so we only report statistics for Dream.

### D.2 Text generation

### D.2.1 Experiment details

We evaluate KL-Adaptive Stability Sampling (KLASS) on a Masked Diffusion Language Model (MDLM) [34] pre-trained on the OpenWebText corpus [13]. As baselines, we include (i) the original autoregressive sampler (one-token-at-a-time unmasking), (ii) SEDD [25], and (iii) two MDLM variants: the standard 512-step sampler and the "absorb" variant [3]. For all diffusion-based methods, we generate 1,000 sequences of length 1,024 tokens under a fixed 512-step schedule, applying nucleus (top-p) filtering at $p = 0.9$, a history length $n = 2$, a KL divergence threshold $\epsilon_{KL} = 1e - 4$, and a confidence threshold $\tau = 0.57$. To ensure a fair comparison under this fixed step count, we cap the maximum number of tokens accepted by the thresholds at each step. In the fallback case where tokens do not pass this criterion, we revert to the original MDLM. We report generative perplexity by exponentiating the average token-level negative log-likelihood under three oracle models (LLaMA2 7B, LLaMA3 8B, and GPT-2), measure Shannon entropy of the predicted token distributions, and compute MAUVE by comparing our 1,000 generated samples to 1,000 held-out segments from the OpenWebText. All runs were executed on a single NVIDIA RTX A6000 GPU. To quantify run-to-run variability, each method was repeated with three fixed random seeds, and we report mean $\pm$ one standard deviation over these replicates.

Table 9: Mean and standard deviation for each sampler across three runs (mean $\pm$ std).

| Sampler | MATH | | GSM8K | |
| | Acc (%) | Step | Acc (%) | Step |
| --- | --- | --- | --- | --- |
| Top-1 | $37.97 \pm 0.12$ | $256.00 \pm 0.00$ | $79.55 \pm 0.14$ | $256.00 \pm 0.00$ |
| Random | $18.73 \pm 1.61$ | $256.00 \pm 0.00$ | $37.35 \pm 0.53$ | $256.00 \pm 0.00$ |
| Top-2 | $33.60 \pm 0.16$ | $128.00 \pm 0.00$ | $71.69 \pm 0.35$ | $128.00 \pm 0.00$ |
| conf $> 0.9$ | $41.80 \pm 0.00$ | $95.10 \pm 0.00$ | $73.67 \pm 0.15$ | $74.81 \pm 0.08$ |
| KL $< 0.001$ | $41.27 \pm 0.09$ | $162.49 \pm 0.00$ | $76.70 \pm 1.14$ | $150.02 \pm 0.32$ |
| KLASS (ours) | $43.20 \pm 0.00$ | $149.72 \pm 0.00$ | $79.43 \pm 0.72$ | $155.67 \pm 0.41$ |

(a) MATH & GSM8K

| Sampler | HumanEval | | MBPP | |
| | Acc (%) | Step | Acc (%) | Step |
| --- | --- | --- | --- | --- |
| Top-1 | $58.53 \pm 0.00$ | $256.00 \pm 0.00$ | $63.81 \pm 0.00$ | $256.00 \pm 0.00$ |
| Random | $18.09 \pm 2.51$ | $256.00 \pm 0.00$ | $28.14 \pm 0.91$ | $256.00 \pm 0.00$ |
| Top-2 | $42.88 \pm 0.29$ | $128.00 \pm 0.00$ | $47.08 \pm 0.32$ | $128.00 \pm 0.00$ |
| conf $> 0.9$ | $50.00 \pm 0.00$ | $52.47 \pm 0.00$ | $57.59 \pm 0.00$ | $72.49 \pm 0.00$ |
| KL $< 0.001$ | $59.35 \pm 0.29$ | $73.94 \pm 0.57$ | $62.65 \pm 0.00$ | $108.15 \pm 0.00$ |
| KLASS (ours) | $59.35 \pm 0.29$ | $74.88 \pm 0.74$ | $64.59 \pm 0.00$ | $111.24 \pm 0.00$ |

(b) HumanEval & MBPP

Table 10: Generative perplexity, MAUVE and entropy on unconditional text generation. Here, 'D3PM' denotes an MDLM that is parameterized using D3PM.

| Method | MAUVE $\uparrow$ | LLaMA2 $\downarrow$ | LLaMA3 $\downarrow$ | GPT2 $\downarrow$ | Entropy $\uparrow$ |
| --- | --- | --- | --- | --- | --- |
| *Data | 1.000 | 7.00 | 9.40 | 14.80 | 5.44 |
| AR | $0.855 \pm 0.033$ | $10.97 \pm 0.10$ | $15.12 \pm 0.18$ | $12.07 \pm 0.12$ | $5.21 \pm 0.02$ |
| SEDD | $0.037 \pm 0.012$ | $53.09 \pm 0.24$ | $109.60 \pm 0.79$ | $105.40 \pm 0.67$ | $5.62 \pm 0.00$ |
| D3PM | $0.022 \pm 0.006$ | $41.82 \pm 5.88$ | $72.85 \pm 12.69$ | $76.70 \pm 0.62$ | $5.40 \pm 0.00$ |
| MDLM | $0.115 \pm 0.033$ | $30.88 \pm 0.20$ | $54.15 \pm 0.27$ | $51.78 \pm 0.14$ | $5.46 \pm 0.00$ |
| KLASS | $\mathbf{0.179 \pm 0.041}$ | $\mathbf{26.94 \pm 0.24}$ | $\mathbf{49.19 \pm 0.40}$ | $\mathbf{45.50 \pm 0.42}$ | $\mathbf{5.43 \pm 0.01}$ |

## D.3 Image generation

### D.3.1 Experiment details

We evaluate KLASS on the MMaDA [49]. We compare two samplers—confidence-based and KLASS. For each sampler, we generate 10,000 class-conditional images with uniformly sampled ImageNet labels under a 16-step, 32-step decoding budget. For KLASS, we fix the hyperparameters to history length $n = 1$, KL divergence threshold $\epsilon_{\mathrm{KL}} = 0.3$, and confidence threshold $\tau = 0.1$. For a fair comparison with a fixed step count, we restrict the per-step reveal count allowed by the thresholds. FID is computed against the 50k ImageNet validation set using official Inception-v3 features, and IS follows the standard protocol. All image-generation runs use a single NVIDIA RTX A5000.

## D.4 Molecules

### D.4.1 Experiment details

We mainly follow the experimental settings in [37] for the experiment. For training, we train independent models for two target conditions: QED and ring count. We use diffusion step size 32, taking 25,000 gradient steps. Small size diffusion transformer (DIT) which is composed of 12 DIT

blocks, hidden dimension of 768 is utilized for the architecture. We train the model with classifier-free guidance (CFG) training with dropout condition probability of 0.1. We generate samples with CFG strength $\gamma = 1$ for the experiment. Reported values in Table 4 for KLASS use $\epsilon_{KL} = 0.001$ and $\tau = 0.98$ for both QED and Ring Count experiments. We utilize a single RTX 3090 GPU for both training and the inference.

## D.5 Ablations

### D.5.1 Hyperparameter Sensitivity

To address sensitivity, we performed a grid search across diverse models and tasks (LLaDA/Dream for reasoning, MDLM for molecular), demonstrating the robustness of KLASS.

Our findings show that configurations near the selected optimum consistently yield high accuracy while significantly reducing sampling steps. For instance, on HumanEval with LLaDA (Table 11), settings near (confidence = 0.9, KL = 0.01) maintain or improve upon the baseline accuracy of **39.63%** while using far fewer than 256 steps. Similar trends are observed for other settings (Tables 12, 13, 14, 15). Slightly different hyperparameter settings can sometimes outperform the main reported configuration, indicating both robustness and potential for further tuning.

### D.5.2 Effect of history length

We evaluate the effect of varying the KL score history length in KLASS across different KL divergence thresholds and confidence thresholds on the MATH dataset. Results are reported in Table 16 for both LLaDA and Dream models.

For LLaDA, a history length of 2 offers the best balance of accuracy and efficiency, particularly at KL threshold of 0.015 and a confidence threshold of 0.6. At the stricter threshold of 0.9, history length has less impact, suggesting that a more relaxed confidence threshold allows more informative token candidates to be considered for unmasking. For Dream, the highest accuracy is achieved with history length 2, $\epsilon_{KL} = 0.005$, and $\tau = 0.9$. At a lower confidence of 0.6, overall accuracy decreases, and longer history helps stabilize token predictions. In summary, history length 2 is optimal across most settings, providing improved accuracy with moderate computational cost. Therefore, we use history length 2 for all reasoning tasks.

### D.5.3 Effect of temperature

Table 17 shows that KLASS consistently improves over a deterministic Top-1 sampler across tasks and temperature settings. Gains are especially strong at temperature 0, where KLASS boosts accuracy by 6.22 to 8.00 percentage points and reduces steps by up to 79%. At temperature 0.2, it still provides solid improvements, with accuracy gains of 0.69 to 5.10 points and step reductions of 39% to 71%. These results highlight KLASS's ability to accelerate reasoning and improve accuracy, with even greater boosts in more deterministic settings.

## E Comparison to other diffusion samplers

### E.1 Performance on reasoning tasks

To recap the baselines used in the main experiment (Table 1), we consider:

- **Top-k**: Tokens are generated by selecting the one with the highest confidence [7].
- **Random**: Tokens are generated in a purely random order [1].

Table 18 reports results for two additional samplers:

- **Top-k Margin**: Unmasks the token with the largest probability margin between the highest and second-highest confidence [21].
- **Entropy**: Tokens are ranked by their negative Shannon entropy, prioritizing those with lower uncertainty (i.e., higher confidence) in the model's prediction.

Table 11: Hyperparameter sensitivity on HumanEval with LLaDA. We report values as Accuracy (Steps). Conf = 0.9 and KL = 0.01 are chosen for KLASS. The baseline accuracy is 39.63% with 256 steps.

|  | KL = 0.015 | **KL = 0.01** | KL = 0.005 |
|---|---|---|---|
| Conf = 0.95 | 39.63 (65.19) | 40.24 (99.29) | 40.24 (103.14) |
| **Conf = 0.9** | 40.24 (89.29) | **40.85 (91.98)** | 40.24 (96.75) |
| Conf = 0.85 | 39.63 (84.98) | 39.63 (88.78) | 40.24 (94.48) |
| Conf = 0.8 | 39.02 (83.14) | 39.02 (87.21) | 40.85 (94.01) |

Table 12: Hyperparameter sensitivity on MBPP with LLaDA. We report values as Accuracy (Steps). Conf = 0.7 and KL = 0.01 are chosen for KLASS. The baseline accuracy is 48.64% with 256 steps.

|  | KL = 0.015 | **KL = 0.01** | KL = 0.005 |
|---|---|---|---|
| Conf = 0.8 | 49.42 (122.37) | 49.42 (134.52) | 49.42 (134.52) |
| Conf = 0.75 | 49.03 (118.61) | 49.42 (123.45) | 48.25 (132.11) |
| **Conf = 0.7** | 48.25 (116.24) | **49.03 (127.81)** | 49.03 (130.67) |
| Conf = 0.65 | 46.30 (113.22) | 48.64 (118.54) | 49.42 (128.99) |

Table 13: Hyperparameter sensitivity on HumanEval with Dream. We report values as Accuracy (Steps). Conf = 0.8 and KL = 0.001 are chosen for KLASS. The baseline accuracy is 58.53% with 256 steps.

|  | KL = 0.005 | KL = 0.003 | **KL = 0.001** | KL = 0.0005 |
|---|---|---|---|---|
| Conf = 0.9 | 54.87 (64.78) | 56.10 (67.41) | 57.93 (74.86) | 61.59 (79.57) |
| Conf = 0.85 | 57.32 (59.39) | 55.49 (65.31) | 59.15 (73.38) | 60.98 (79.43) |
| **Conf = 0.8** | 51.22 (58.09) | 54.27 (62.82) | **59.76 (73.73)** | 60.96 (79.51) |
| Conf = 0.75 | 48.78 (55.21) | 53.05 (62.61) | 59.15 (73.41) | 60.37 (79.37) |

Table 14: Hyperparameter sensitivity on MBPP with Dream. We report values as Accuracy (Steps). Conf = 0.9 and KL = 0.001 are chosen for KLASS. The baseline accuracy is 63.81% with 256 steps.

|  | KL = 0.005 | KL = 0.003 | **KL = 0.001** | KL = 0.0005 |
|---|---|---|---|---|
| Conf = 0.95 | 65.37 (108.93) | 65.37 (110.03) | 64.59 (112.56) | 64.59 (113.14) |
| **Conf = 0.9** | 62.65 (103.99) | 64.20 (107.09) | **64.59 (111.24)** | 64.59 (112.65) |
| Conf = 0.85 | 64.20 (101.34) | 63.81 (105.43) | 65.37 (112.54) | 64.59 (112.76) |
| Conf = 0.8 | 63.81 (96.02) | 63.04 (103.22) | 65.37 (112.52) | 64.59 (112.82) |

Table 15: Hyperparameter sensitivity on Molecule QED (MDLM). Conf = 0.98 and KL = 0.001 are chosen for KLASS. The baseline QED is 0.526 with 32 steps. We report values as QED (Steps).

|  | KL = 0.01 | KL = 0.005 | **KL = 0.001** | KL = 0.0005 |
|---|---|---|---|---|
| Conf = 0.999 | 0.527 (18.61) | 0.526 (18.61) | 0.524 (18.58) | 0.538 (18.45) |
| Conf = 0.99 | 0.515 (18.22) | 0.521 (18.33) | 0.543 (18.69) | 0.537 (18.66) |
| **Conf = 0.98** | 0.531 (18.43) | 0.517 (18.38) | **0.546 (18.78)** | 0.534 (18.82) |
| Conf = 0.96 | 0.529 (18.43) | 0.535 (18.51) | 0.534 (18.63) | 0.543 (18.80) |

KLASS consistently outperforms Top-k Margin and Entropy-based methods with fewer sampling steps. On LLaDA, it achieves top results on GSM8K and HumanEval in less than half the usual iterations. While Entropy yields the highest accuracy on MATH and MBPP with 256 steps, its performance drops sharply with fewer steps. In contrast, KLASS maintains high accuracy at lower computational cost. On Dream, it also improves MATH and GSM8K accuracy while reducing steps, demonstrating more efficient and effective sampling.

Table 16: Ablation results on the history length, across KL thresholds and models, grouped by confidence thresholds.

| Conf | KL | History Length | LLaDA Acc (%) | LLaDA Steps | Dream Acc (%) | Dream Steps |
|---|---|---|---|---|---|---|
| 0.6 | 0.010 | 1 | 32.2 | 77.13 | 37.6 | 123.73 |
| | | 2 | **33.8** | 128.62 | 39.6 | 165.68 |
| | | 3 | 32.2 | 153.60 | 41.8 | 182.93 |
| | 0.015 | 1 | 30.4 | 89.04 | 34.8 | 83.11 |
| | | 2 | 33.2 | 121.11 | 35.6 | 119.33 |
| | | 3 | 33.2 | 146.19 | 36.6 | 146.40 |
| | 0.020 | 1 | 31.0 | 74.42 | 34.4 | 73.66 |
| | | 2 | 31.0 | 117.01 | 29.6 | 104.19 |
| | | 3 | 30.6 | 140.56 | 34.6 | 128.46 |
| 0.9 | 0.010 | 1 | 31.4 | 128.76 | 42.2 | 141.36 |
| | | 2 | 30.6 | 152.75 | 41.6 | 169.45 |
| | | 3 | 30.8 | 170.66 | 42.0 | 183.51 |
| | 0.015 | 1 | 31.0 | 127.05 | 41.0 | 126.42 |
| | | 2 | 30.8 | 149.72 | **43.2** | 149.72 |
| | | 3 | 31.0 | 167.11 | 40.2 | 165.37 |
| | 0.020 | 1 | 30.8 | 125.85 | 40.4 | 120.35 |
| | | 2 | 30.8 | 148.92 | 42.0 | 142.75 |
| | | 3 | 31.2 | 164.75 | 40.4 | 158.01 |

Table 17: Effect of temperature on KLASS gains over Top-1 sampler with Dream.

| Method | MATH Acc | MATH Steps | GSM8K Acc | GSM8K Steps | HumanEval Acc | HumanEval Steps | MBPP Acc | MBPP Steps |
|---|---|---|---|---|---|---|---|---|
| | | | | Temperature = 0.2 | | | | |
| Top-1 | 38.10 | 256 | 79.75 | 256 | 58.53 | 256 | 63.81 | 256 |
| KLASS | 43.20 $_{+5.10}$ | 150 $_{-106}$ | 80.44 $_{+0.69}$ | 156 $_{-100}$ | 59.76 $_{+1.23}$ | 74 $_{-182}$ | 64.59 $_{+0.78}$ | 111 $_{-145}$ |
| | | | | Temperature = 0 | | | | |
| Top-1 | 25.80 | 256 | 41.70 | 256 | 29.27 | 256 | 33.46 | 256 |
| KLASS | 33.80 $_{+8.00}$ | 121 $_{-135}$ | 47.92 $_{+6.22}$ | 106 $_{-150}$ | 37.19 $_{+7.92}$ | 53 $_{-203}$ | 40.86 $_{+7.40}$ | 76 $_{-180}$ |

# F  Examples of generated samples

We present a qualitative comparison of generated samples to demonstrate KLASS's improvements. Table 19 illustrates a mathematical reasoning task. The Top-1 baseline fails due to a simple arithmetic error in the first step ($f(-2) = \frac{-8}{2} = -4$), while the random sampling disregards the function's structure. In contrast, KLASS correctly computes each intermediate step and combines them to reach the correct answer, $\frac{14}{3}$, highlighting its improved reasoning reliability. We also compare long-form text coherence. Figure 4 shows an MDLM baseline sample that begins on-topic ('SolarCity') but quickly degenerates. It exhibits severe generation artifacts such as repetition ('SolarCitySolarCity'), nonsensical phrases ('informs of capacity'). The sample's coherence completely breaks down, ending with an unrelated spam link. Conversely, Figure 5 shows a KLASS sample on 'urban sprawl.' This text is relatively coherent and topically consistent from start to finish, maintaining a plausible journalistic style.

Table 18: Performance and sampling steps on reasoning benchmarks for Top-k Margin and Entropy samplers.

| Model | Method | MATH Acc↑ | MATH Step↓ | GSM8K Acc↑ | GSM8K Step↓ | HumanEval Acc↑ | HumanEval Step↓ | MBPP Acc↑ | MBPP Step↓ |
|-------|--------|-----------|------------|------------|-------------|----------------|-----------------|-----------|------------|
| LLaDA | Top-k Margin | 32.0 | 256 | 74.14 | 256 | 39.63 | 256 | 47.86 | 256 |
|  | Top-k Margin | 31.4 | 128 | 74.45 | 128 | 30.48 | 128 | 40.08 | 128 |
|  | Entropy | **34.6** | 256 | 75.43 | 256 | 35.97 | 256 | **51.75** | 256 |
|  | Entropy | 32.6 | 128 | 73.01 | 128 | 25.60 | 128 | 40.08 | 128 |
|  | KLASS (ours) | 33.8 | 128.62 | **76.50** | 98.57 | **40.85** | 91.98 | 47.86 | 119.59 |
| Dream | Top-k Margin | 39.4 | 256 | 79.45 | 256 | 58.53 | 256 | 63.81 | 256 |
|  | Top-k Margin | 32.4 | 128 | 71.49 | 128 | 43.29 | 128 | 46.69 | 128 |
|  | Entropy | 39.4 | 256 | 79.45 | 256 | 58.53 | 256 | 63.81 | 256 |
|  | Entropy | 32.6 | 128 | 71.49 | 128 | 43.29 | 128 | 46.69 | 128 |
|  | KLASS (ours) | **43.2** | 149.72 | **80.44** | 156.24 | **59.76** | 73.73 | **64.59** | 111.24 |

# G  Limitation & Broader impact

## G.1  Limitations

**Further scalability to larger models**  In the absence of larger-size discrete diffusion models compared to AR models, our method cannot be evaluated on the more challenging benchmarks such as in agentic systems of LLMs [22, 39, 44].

**Hyperparameter Search Cost**  Although we provide a lightweight guideline in Appendix D.1.2, KLASS introduces a minimal search cost for its additional hyperparameters compared to simpler baselines like Top-1 sampling. While our analysis in Appendix D.5.1 shows performance is robust around the optimal points, further tuning could still be performed to find the best possible settings.

## G.2  Broader impact

This work focuses on how to improve discrete diffusion sampler. However, improving diffusion samplers, if misused, could facilitate the creation of harmful or misleading text, images, or even novel molecules for malicious purposes.

To mitigate these risks, improvements in sampling techniques should be accompanied by a strong emphasis on AI safety research and ethical guidelines. This includes developing robust mechanisms for detecting and preventing the generation of harmful content and fostering public discourse on the societal implications of advanced generative models.

Table 19: Generated solutions with Dream on MATH.

| Question | If $f(x) = \frac{3x-2}{x-2}$, what is the value of $f(-2) + f(-1) + f(0)$? Express your answer as a common fraction. |
|---|---|
| **Answer** | $\frac{14}{3}$ |
| **Top-1** | First, let's find $f(-2)$: $f(-2) = \frac{3(-2)-2}{-2-2} = \frac{-8}{2} = -4$ Next, let's find $f(-1)$: $f(-1) = \frac{3(-1)-2}{-1-2} = \frac{-5}{-3} = \frac{5}{3}$ Now, let's find $f(0)$: $f(0) = \frac{3(0)-2}{0-2} = \frac{-2}{-2} = 1$ Finally, add: $f(-2) + f(-1) + f(0) = -4 + \frac{5}{3} + 1 = -4 + \frac{8}{3} = \frac{-12}{3} + \frac{8}{3} = \frac{-4}{3}$ The final answer is $\frac{-4}{3}$. |
| **Random** | $f(-2) = \frac{3(-2)}{-2-2} = 1$, $f(-1) = \frac{3(-2)}{-1-2} = 1$, $f(0) = \frac{3(-2)}{0-2} = 1$. So $f(-2) + f(-1) + f(0) = 3$. The final answer is 3. |
| **KLASS** | First, let's find $f(-2)$: $f(-2) = \frac{3(-2)-2}{-2-2} = \frac{-8}{-4} = 2$ Next, let's find $f(-1)$: $f(-1) = \frac{3(-1)-2}{-1-2} = \frac{-5}{-3} = \frac{5}{3}$ Now, let's find $f(0)$: $f(0) = \frac{3(0)-2}{0-2} = \frac{-2}{-2} = 1$ Finally, add: $f(-2) + f(-1) + f(0) = 2 + \frac{5}{3} + 1 = 2 + \frac{8}{3} = \frac{6}{3} + \frac{8}{3} = \frac{14}{3}$ The final answer is $\frac{14}{3}$. |

Figure 4: MDLM Generated Sample (512 tokens)

```
<endoftext>City\u2019s new Solar City facility in San Diego will be
providing up to 4 times New York\u2019s current capacity as energy
efficient storage is on the way, according to new reporting
StarTheSunCity reports.\n\nSolarCity says a site in Los Angeles is
designed to replace NYC\u2019s current capacity in the emergency room. As
 SolarCity Krieger confirmed, a facility in Richmond is planned for
SolarCitySolarCity.\n\nThe company plans to adopt the city\u2019s own
capacity taking advantage that the city is constantly being built to low
energy standards.
...
The NYTChronicle reports that SolarCity is aiming to provide a capacity
increase by 10 to 16 percent of capacity, greater than New York. Even the
 non-Manhattan capacity increase is about 2 percent.\n\nStarTheSunCity
says it will soon run out of space for an increasing number of 'informs of
capacity'\u2019 including some rooms not dictated by capacity requirements
.\n\nThe indoor solar center will be built to pay solar customers a $3.40
 price per day, and based according to the budget for
SolarCitySanVoltSolarCity\u2019s facility opens in 2017, and the City Center,
 located in just one block from Microsoft Place, seats 25 million New
Yorkers. SolarCity spokesman Barry said, \u201cIt is clear that the city
represents the best leaders in energy efficiency. In fact, providing so
much more energy for people is a vital leadership position for consumers\
u2019 transportation, health, and finances over the next decade.\u201d\n\
nClick to Download The New York Charger and look for True New York
Charger to feel free download or for support.<endoftext>
```

Figure 5: KLASS Generated Sample (512 tokens)

```
<|endoftext|> an order of magnitude smaller than a rise in the number of
people living in other cities.\n\nAs a result, urban sprawl is now
central to a realignment in how planners view the contribution of each
individual population and location to different cities.\n\nThe University
 researchers measured growth in densities by a factor of two and space
and pollution by 31 per cent in the period of urban sprawl more broadly.\
n\nThere is, of course, a period of growth in the number of inhabitants
relocated or public space by a quarter to two per cent.
...
Again, at this volume, it has tended to grow unevenly in other cities.\n\
nThe Berlin Centre for Urban Research has a map of low urban sprawl areas
 around the borders in Brussels, Belgium.\n\nLichtner and co-authors of
the study, published in UrbanLabs West, say places with more intense
urban sprawl need infrastructure to expand public space.\n\nAs a result,
their boundaries are denser, that means more land for construction and
more roads for the new developers want to build. Public libraries and
classrooms also go up dramatically, supposedly making urban sprawl more
severe and steeper.\n\nLichtner and co-authors say the foundations for
urban sprawl are powerful in this respect. They say much of the impact of
 the study is ultimately centred on the urban sprawl itself, that offers
the means to thrive. \"If sprawl is the city, it might be infrastructure
that allows for housing in sprawl, creating a productive concentration
between the wealthier and the poor that stand for a new development,\"
they said.<|endoftext|>
```

