# OpenReview forum: "KLASS: KL-Guided Fast Inference in Masked Diffusion Models"
_NeurIPS.cc/2025/Conference — NeurIPS 2025 spotlight_

### Official Review · Reviewer_5Uk3 · 2025-06-25

**Clarity:** 3
**Significance:** 4
**Originality:** 4
**Rating:** 6
**Confidence:** 3

**Summary:**

The authors tackle the well-known latency problem of discrete/masked diffusion models by proposing KL-Adaptive Stability Sampling (KLASS). In every reverse-diffusion step the model already produces a full categorical distribution for each still-masked position. KLASS exploits this "free" information in two ways:

1.Token-level confidence (max p).


2. Inter-step stability (KL divergence between the current and previous logits).

A position whose confidence exceeds τ and whose KL trajectory stays below ε for n recent steps is deemed stable and is unmasked immediately. If no position meets the twin criteria, KLASS falls back to unmasking the top-u most-confident tokens. Because many positions converge early, the total number of sampling steps roughly halves, giving ≈2× wall-clock speed-ups without additional training or auxiliary models.

The paper supplies:

1. Theory (Theorem 4.3) arguing that low-KL tokens are (near) optimal under the true data distribution, and a cost analysis (Prop 4.4).

2. Experiments on four reasoning benchmarks, unconditional text generation, ImageNet 256×256, and QM9 molecule design. Gains of +1–3 pp accuracy (Math, GSM8K, HumanEval) with ≈50 % fewer steps, lower text perplexity and higher MAUVE, FID↓ and IS↑ on images, and higher QED for molecules are reported.

3. A detailed checklist and an openly released, anonymised code package.

The proposal is simple, training-free, and broadly applicable.

**Questions:**

Will we see this made available via a GUI or code repo?

**Ethical Concerns:**

["NO or VERY MINOR ethics concerns only"]

**Final Justification:**

Excellent quality rebuttal by the authors. This is strong work, and sampling improvements help the whole field. We need more papers like this.

**Quality:**

4

**Strengths And Weaknesses:**

Strengths

The paper addresses slow inference in masked diffusion models and proposes a delightfully simple yet powerful remedy. By harvesting information the model already computes (per-token confidence and inter-step KL divergence), KL-Adaptive Stability Sampling (KLASS) doubles generation speed without retraining or auxiliary networks. The empirical study is unusually broad: the authors test on four reasoning benchmarks, open-domain text generation, ImageNet-scale image synthesis, and QM9 molecular design, and in every case KLASS maintains or improves quality while cutting steps roughly in half. The ablation analyses are informative, showing how confidence and KL interact and why multi-token unmasking is important. Reproducibility is also taken seriously. Hyper-parameters, compute budgets, and anonymised code are all supplied which makes it straightforward for the community to verify and build upon the work. Finally, the contribution is conceptually elegant. It extracts extra mileage from the model’s own logits instead of bolting on planners or performing expensive distillation, a design that is likely to be adopted quickly by practitioners.

Weaknesses

Despite those merits, several issues temper the enthusiasm. The theoretical underpinning (Theorem 4.3) is sketched at a high level and leaves key quantities such as the size of the "mild context change" and the precise δ-approximation norm undefined, so the guarantee feels more heuristic than rigorous. The results table reports only the number of diffusion steps, not actual wall-clock latency -  without single-query timings it is hard to gauge the practical speed-up!!!! The reasoning benchmarks are presented as single scores with no variance estimates, leaving open the question of statistical significance for the modest (+1 pp) improvements on GSM8K. Parameter sensitivity is explored only for math tasks, so the assertion that one set of ε and τ works universally remains unsubstantiated, and the memory/time overhead of storing KL histories for thousand-token sequences is acknowledged only qualitatively. Finally, the draft’s presentation could benefit from polishing (minor typos were found).

---

> ### Author Rebuttal · Authors · 2025-07-31
>
> We sincerely thank you for your comprehensive and positive comments on our manuscript. We are especially grateful for your recognition of our key contributions:
>
> - The conceptual elegance and practical power of our simple, training-free method.
> - The strength and breadth of our empirical study, demonstrating consistent quality improvements and significant speed-ups across diverse domains.
> - Its reproducibility, ensured by our provision of code and detailed experimental settings.
>
> We have carefully considered your suggestions for improvement and provide our detailed responses below.
>
> ---
> ### [W4-1] Theoretical Guarantee
>
> We thank the reviewer for their helpful comments on Theorem 4.3. We agree that its primary role was to build intuition. Motivated by your feedback, we have developed a new, rigorous information-theoretic framework that formally proves the effectiveness of KLASS. This now serves as the primary theoretical foundation in our revised manuscript.
>
> The primary goal of the original Theorem 4.3 was to provide a high-level intuition for why KLASS is effective. Our intended definitions were as follows:
>
> * **"Mild context change"**: We intended this to mean a change between two contexts, $x_{-i}$ and $x'_{-i}$, where the Hamming distance is bounded by a small integer $k$. This is empirically grounded, as standard samplers like Top-1 or Top-2 correspond to k=1 or k=2. Our own adaptive method, KLASS, also results in a small average k (typically 2-4 tokens per step), confirming the changes are indeed mild.
>
> * **"δ-approximation norm"**: This refers to the distance between our model's distribution, $p_\theta$, and an ideal target distribution, $\pi$, as measured by KL Divergence. Formally, the assumption is that the model's approximation error is bounded such that $D_{KL}(\pi \| p_\theta) \le \delta$.
>
> Formalizing these concepts as you suggested indeed makes the original theorem's structure more rigorous. Taking your valuable feedback a step further, we were motivated to develop an entirely new proof that provides a more direct and quantitatively verifiable guarantee, which we believe better solidifies the theoretical foundations of our work.
>
>
> #### **Provable Framework for Token Unmasking Safety**
>
> To complement this high-level intuition with a rigorous proof, we have developed a new, self-contained theoretical framework built on standard principles of information theory. This new information-theoretic approach directly analyzes the risk of unmasking a stable token. It provides a **provable, quantitative upper bound** on this risk, thereby offering a formal and rigorous complement to the high-level intuition provided by our original theorem.
>
> To make our new framework clear, we first define the key terms used in our theorem:
> * **Risk Measure (CMI), $I_t(i)$**: We quantify the risk of unmasking token $i$ at step $t$ using the Conditional Mutual Information, $I_t(i) = I(X_i; Z \mid V_t)$, which measures its dependency on all other masked tokens $Z$. A low CMI means the token is "safe" to unmask.
> * **Cone Entry Step, $t_\star(i)$**: This is the first step where token $i$ becomes 'stable' by satisfying both KL stability ($K_{t,n}(i) \leq \varepsilon$) and confidence ($\max_v p_t(X_i=v) \geq \tau$) thresholds.
> * **Post-Stability KL Tail, $\delta_t(i)$**: This term aggregates the small, confidence-weighted KL divergences ($\Delta$) of a token *after* it has entered the stability cone:
>     $$
>     \delta_t(i) = \sum_{k=t_\star(i)}^{t-1} [1-c_{k+1}(i)]\Delta_{k+1}(i)
>     $$
> * **Initial Entropy, $H_{t_\star(i)}(X_i)$**: This is the uncertainty (entropy) of token $i$ at the precise moment it becomes stable. Due to the high confidence requirement for entering the cone, this value is provably small.
>
> **Theorem (Provable Bound for Token Safety)**
> Let $i$ be a token that is inside the stability cone at step $t$, and let $t_\star(i)$ be the step at which it first entered the cone. The mutual information $I_t(i)$ is bounded by:
> $$
> I_t(i) \leq \Gamma_t(i) := \delta_t(i) + H_{t_\star(i)}(X_i)
> $$where $\delta_t(i)$ is the small(empirically lower than 1bits), accumulated KL divergence after stabilization and $H_{t_\star(i)}(X_i)$ is the token's entropy at the moment it entered the cone.
>
> **Corollaries: Pairwise and Block MI Bounds**
> For any set of stable tokens $\mathcal{S}_t = \{ i_1, \dots, i_m \}$ unmasked in parallel at step $t$, the total dependency within the block is bounded by the sum of individual risks:
>
> $$I(X_{i_1}; \dots; X_{i_m} \mid V_t) \le \sum_{k=1}^{m} \Gamma_t(i_k)$$
>
> #### **How the New Theory Justifies KLASS**
> Our new theory provides a rigorous justification for KLASS's core strategy of safe, parallel unmasking. We formalize the dependency of a candidate token on other unknowns using Conditional Mutual Information $I_t(i)$ and prove that KLASS's criteria guarantee this value is bounded by a tiny amount $\Gamma_t(i)$. Empirically, we measured this bound to be as low as 0.6 bits on GSM8K, confirming that a token's distributional stability is a strong indicator of its informational independence.
>
> This guarantee extends to the multi-token case. Our corollaries show that the total dependency within an entire block of stable tokens also remains minimal, formally validating why KLASS's parallel decoding is not just fast, but also safe.
>
> ### [W4-2] Lack of Wall-Clock Latency
> We provide detailed wall-clock latency comparisons between standard top-1 confidence decoding and KLASS across multiple datasets and models. As shown below, KLASS achieves consistent speedups-up to 2.78× faster—while maintaining or improving accuracy. This highlights its practical benefits for reducing decoding time without sacrificing performance.
>
> | **Model** | **Dataset** | **Top-1 Acc** | **KLASS Acc** | **Top-1 Time (s)** | **KLASS Time (s)** | **Speedup** |
> | --- | --- | --- | --- | --- | --- | --- |
> | **LLaDA** | GSM8K | 75.13 | 76.50 | 37.04 | 15.86 | **2.34x** |
> | **LLaDA** | MATH | 31.40 | 33.80 | 38.40 | 21.41 | **1.79x** |
> | **LLaDA** | Humaneval | 39.63 | 40.85 | 39.54 | 16.04 | **2.47x** |
> | **LLaDA** | MBPP | 46.69 | 47.86 | 39.12 | 20.68 | **1.89x** |
> | **Dream** | GSM8K | 79.75 | 80.44 | 29.66 | 22.26 | **1.33x** |
> | **Dream** | MATH | 38.00 | 43.20 | 30.76 | 23.31 | **1.32x** |
> | **Dream** | Humaneval | 58.53 | 59.76 | 32.01 | 11.52 | **2.78x** |
> | **Dream** | MBPP | 63.81 | 64.59 | 31.89 | 17.65 | **1.81x** |
>
> ### [W4-3] Uncertainty in GSM8K Improvement Significance
> While LLaDA results are deterministic, Dream involves sampling with temperature = 0.2, following the default settings used in prior work. Accordingly, we report variance estimates for Dream, computed over three independent runs for each method, across all four reasoning benchmarks in Table 9 of the appendix.
>
> On GSM8K, we acknowledge that the accuracy improvement is modest and not statistically significant. However, KLASS achieves a substantial reduction in decoding steps (e.g., from 256 to 155.67), offering meaningful efficiency gains. Additionally, the accuracy improvements and step reduction on other tasks are statistically significant and consistent, reinforcing the overall effectiveness of KLASS across diverse reasoning benchmarks.
>
> ### [W4-4] Unsubstantiated Universal Parameter Sensitivity
> We thank the reviewer for raising this point. Our results indicate that performance is **robust to hyperparameter choices**, remaining stable across different settings. Full grid search results are provided in **\[W1-1]**.
>
> While optimal hyperparameters may vary slightly across tasks, we also provide a **practical hyperparameter search guideline** requiring only 100 validation samples:
>
> 1. **KL Initialization**: Set the KL threshold to the median per-token KL divergence on a small validation set.
> 2. **Confidence Sweep**: With KL fixed, sweep the confidence threshold to balance accuracy and decoding cost.
> 3. **KL Refinement**: With confidence fixed, fine-tune the KL threshold around the initial value for further gains.
>
> In **[W1-1]**, we report a local sensitivity analysis guided by this procedure. Across all tasks, nearby configurations match or fall within 1% of the best accuracy while maintaining similar decoding efficiency. This confirms that our method is robust and exhibits low sensitivity to hyperparameter choices.
>
> ### [W4-5] Qualitative KL History Overhead
> We thank the reviewer for raising the concern about memory and time overhead for longer sequences. In practice, this overhead is very small, as KLASS is a lightweight post-processing step over existing logits and requires no additional forward pass.
> For masked tokens $I_m = \lbrace i \mid z^i_t = m \rbrace$, KLASS computes KL scores $d^i_t = D_{\mathrm{KL}}(p^i_t \,\|\, p^i_{t-1})$ and caches the prior distributions, resulting in a compute and memory cost of $O(|I_m| \cdot |V|)$. These vector operations are negligible compared to the expensive matrix multiplications and memory usage of the diffusion process.
>
> Empirically, even for sequences up to 1,024 tokens, our wall-clock measurements show **<5% memory overhead** and **<0.4% time overhead** per decoding step, as shown below:
>
> |Model|Vocab Size|Length|Memory Overhead (MB)|Total Memory (MB)|Time Overhead (s)|Total Time per Step (s)|
> |---|---|---|---|---|---|---|
> |**LLaDA**|126,464|256|247|18,702|0.000255|0.1218|
> |**LLaDA**|126,464|512|494|19,996|0.000673|0.1889|
> |**LLaDA**|126,464|1024|988|22,589|0.00108|0.3114|
> |**Dream**|152,064|256|296|18,875|0.000177|0.1275|
> |**Dream**|152,064|512|592|20,801|0.000272|0.1852|
> |**Dream**|152,064|1024|1184|24,663|0.000515|0.2968|
>
> ### [W4-6] Requiring Presentation Refinement
> We will thoroughly proofread and polish the final version, addressing all minor typographical errors and enhancing overall clarity.
>
> ### [Q4-1] Code Releasing
>
> Yes, we will make the code publicly available in an open-source repository to encourage further research and use by the community.

---

> > ### Comment · Reviewer_5Uk3 · 2025-08-05
> > **Excellent Rebuttal, I am raising my score.**
> >
> > In light of the significant work done by the authors, I have raised my score to a "6" - Strong accept, and raised my subscores accordingly.
> >
> > Excellent work to all involved! Very pleasing to see further innovation in the sampling space!

---

> > > ### Author Response · Authors · 2025-08-07
> > >
> > > Dear reviewer 5Uk3,
> > >
> > > Thank you very much for your generous feedback and for raising your score to a strong accept. Your constructive comments have been extremely helpful in strengthening our work. We are grateful for your support and look forward to further developing this line of research and sharing our future progress with the community.

---

### Official Review · Reviewer_yKia · 2025-06-25

**Clarity:** 3
**Significance:** 2
**Originality:** 3
**Rating:** 4
**Confidence:** 4

**Summary:**

This paper introduces ‘KL-Adaptive Stability Sampling’ (KLASS), a fast sampling method that exploits token-level KL divergence to identify stable, high-confidence predictions. This paper also theoretically analyzes why ‘KLASS’ improves sample quality in certain scenarios.

**Questions:**

Under mask based generation, the total sampling steps are usually predefined, for example, by setting the cosine scheduler. Under the proposed threshold strategy, it seems that the generated tokens at each step is uncertain. Then, how to define the final total sampling steps.

**Ethical Concerns:**

["NO or VERY MINOR ethics concerns only"]

**Final Justification:**

Most of my concerns are addressed. I lean towards raise my score.

**Limitations:**

Please refer to the weakness part.

**Quality:**

3

**Strengths And Weaknesses:**

Strengths:
1. The proposed low-KL as stabe token is rational.
2. The theoretical analyses are presented on why ‘KLASS’ improves sample quality in certain scenarios.
3. The experiemntal results effectively verified the effectiveness.

Weaknesses:
1. What is the relationship between KL metric and confidence. It seems that these two metrics largely overlap on most of the predicted tokens. Given that confidence metric is a basic strategy, the relationship between these two metrics is important.
2. Compared to the original confidence setting in MaskGIT，the proposed KL metric brings very marginal performance lift.
3. The hyperparameter selection of threshold varies across tasks and models, limiting the practicality.

---

> ### Author Rebuttal · Authors · 2025-07-31
>
> We are grateful for your detailed and constructive feedback. We were pleased to see you recognize that our work:
>
> - Rests on a rational principle supported by theoretical analysis.
>
> - Validates its effectiveness with comprehensive experimental results.
>
> We have carefully considered your comments and provide our detailed responses below.
>
> ---
>
> ### [W3-1] Relationship between KL metric and confidence
> We agree their relationship is critical. Below, we clarify its relationship.
>
> **1. Correlation between confidence and KL metric**
>
> To better understand the relationship between confidence and KL divergence, we measured the Pearson correlation coefficient between the two metric. We conducted this analysis by recording the confidence and KL divergence of the decoded tokens. As shown in the table below, the correlation between these two metrics is relatively low across all datasets. This indicates that the confidence and KL metrics capture different aspects of model behavior, thus making them complementary rather than overlapping.
> | Model     | GSM8K  | MATH   | HumanEval | MBPP   |
> | --------- | ------ | ------ | --------- | ------ |
> | **LLaDA** | -0.092 | -0.174 | -0.213    | -0.187 |
> | **Dream** | -0.247 | -0.253 | -0.306    | -0.297 |
>
> **2. Impact of using only one metric**
>
>  As shown in Table 1 of our manuscript, when we rely on either confidence or KL divergence alone, the model's performance drops compared to when both metrics are utilized together. This supports the argument that while each metric has its own utility, their combined use yields a more robust decision-making process.
>
> **3. Semantic Distinction**
>
> Our empirical and theoretical analyses confirm that confidence and KL divergence are complementary, not redundant. Their Pearson correlation is consistently low across all datasets, and as we show in response **[W2-1]**, correctly generated tokens exhibit both higher confidence and lower KL divergence. This motivates their combined use, as they capture fundamentally different aspects of the model's prediction process.
>
> Confidence provides a static snapshot of certainty at a single timestep ($p_\theta(x_{t, \ell} | z_t)$). It measures the peak of a probability distribution, but it contains no information about the iterative process that led to that state. Relying on confidence alone can lead to errors from premature commitments to transient spikes of certainty.
>
> Divergence, by contrast, measures the dynamics of the model's predictions over time by comparing consecutive distributions ($p_\theta(\cdot | z_t)$ and $p_\theta(\cdot | z_{t-1})$). A low KL score is a direct signal that the model's iterative refinement has concluded for that token—its thought process has converged and stabilized. It is a measure of temporal consistency.
>
> KLASS requires both a confident state and a stable flow. This two-factor check allows us to fully leverage the diffusion model's iterative refinement capability, ensuring we unmask tokens only when the model's prediction is both certain and has verifiably ceased to fluctuate.
>
> ### [W3-2] Peformance gap with MaskGIT setting
> Thank you for the constructive feedback. While FID score improvement seems marginal compared to the baseline method, significance of KLASS is particularly evident in the Inception Score (IS) [1]. Specifically, KLASS achieved a **9.2-point** increase in IS over the strong baseline (**43% relative gain**) which indicates substantial improvement in sample quality and diversity [2]. By waiting for a prediction to become stable (low KL-divergence) rather than exploiting only confidence value, KLASS enables generating complex classes where confidence-based samplers might often fail, which is indicated by the high IS score. Furthermore, we would like to point out that the reported value has further room for improvement as we conducted only naive hyper-parameter search for the experiment. We will include this discussion in our final manuscript.
>
> ### [W3-3] Hyperparameter selection of threshold
> We fully agree that the practicality of our method hinges on a systematic hyperparameter selection process. To this end, all our experiments were guided by a lightweight, three-step guideline we developed for this purpose. Below, we outline this practical guideline—which requires only a small validation set—and demonstrate its low sensitivity using Dream on MBPP as a concrete example. We kindly ask you to refer **[W1-1]** for additional results on hyperparameter sensitivity.
>
> **Step 1. KL Threshold Estimation**:
> Run baseline decoding on the validation set and collect KL values. Set the initial KL threshold to the median of this distribution (e.g. $\approx$ 0.01 for reasoning tasks).
>
> **Step 2. Confidence Threshold Search**:
> Fix the KL threshold from Step 1. Sweep confidence from 0.9 down to 0.6, selecting the one that maximizes accuracy while maintaining efficiency. For Dream on MBPP, fixing KL = 0.01 yields:
>
> | **Conf** | Acc (%) | NFE    |
> | -------- | ------- | ------ |
> | 0.9      | **67**  | 100.93 |
> | 0.8      | 66      | 89.36  |
> | 0.7      | 64      | 75.03  |
> | 0.6      | 62      | 62.60  |
>
> Both 0.9 and 0.95 tie in accuracy, but confidence = 0.9 uses fewer evaluations. We therefore select conf = 0.9.
>
>
> **Step 3. KL Threshold Refinement**:
> Fix confidence at the value from Step 2 and perform a finer KL sweep around the initial estimate. Results with fixing conf = 0.9 are:
>
> | **KL** | Acc (%) | NFE    |
> | ------ | ------- | ------ |
> | 0.015  | 65      | 98.98  |
> | 0.01   | 67      | 100.93 |
> | 0.005  | 67      | 105.05 |
> | 0.001  | **68**  | 109.47 |
>
> Peak accuracy at KL = 0.001 confirms both the initial estimate and its optimal refinement, giving the final configuration (conf=0.9, kl=0.001).
>
> By optimizing confidence and KL in this order, we converge on strong configurations with far fewer evaluations than a full grid search. Moreover, since thresholded decoding reduces per-sample latency, the search process is fast and practical.
>
> Using the selected configuration, we achieve 64.59% accuracy with 111.24 steps—outperforming the top-confidence baseline (63.81% accuracy with 256 steps). Notably, performance remains stable across nearby settings, demonstrating robustness and low sensitivity to hyperparameter choices. The table below shows results on the full dataset:
>
> **Dream on MBPP**
>
> | **Conf** | KL = 0.005     | KL = 0.003     | **KL = 0.001**     | KL = 0.0005    |
> | -------- | -------------- | -------------- | ------------------ | -------------- |
> | 0.95     | 65.37 (108.93) | 65.37 (110.03) | 64.59 (112.56)     | 64.59 (113.14) |
> | **0.9**  | 62.65 (103.99) | 64.20 (107.09) | **64.59 (111.24)** | 64.59 (112.65) |
> | 0.85     | 64.20 (101.34) | 63.81 (105.43) | 65.37 (112.54)     | 64.59 (112.76) |
> | 0.8      | 63.81 (96.02)  | 63.04 (103.22) | 65.37 (112.52)     | 64.59 (112.82) |
>
> *Each cell: Accuracy (Steps)*
>
> Accuracy stays within 1% of the selected configuration across variations, with potential for even further tuning. This stability, combined with minimal search cost, highlights the practicality and generalizability of our guideline across tasks and models.
>
>
> ### [Q3-1] Final total sampling steps
> Thank you for raising insightful question. KLASS operates within a predefined maximum number of sampling steps but shortens the actual steps used by unmasking a variable number of tokens per step based on the KLASS criterion. If no tokens meet the criterion, the top-k most confident tokens are selected to satisfy a minimum number of unmaskings per step—determined to ensure full sequence completion within the step budget. This strategy allows KLASS to often finish earlier while never exceeding the predefined maximum.
>
> Additionally, KLASS can be configured to match a fixed number of steps exactly by selecting the required number of tokens from those satisfying the KLASS criterion. To enforce a predictable lower bound on the total sampling steps, we may guide the process by the analytically computable expectation of token transitions from the predefined reverse process. At each step, we decode all tokens that meet the KLASS criterion and then fill any deficit with top-k decoding to meet the target number for that step.
>
> We have added further results demonstrating that our method also performs well under a shorter predefined budget of 128 steps. While respecting this maximum, KLASS often completes generation in fewer steps and achieves higher accuracy than the top-k confidence baseline.
>
> |**Model**|**Method**|**GSM8K**|**MATH**|**HumanEval**|**MBPP**|
> |---|---|---|---|---|---|
> |**LLaDA**|Top-k|72.40 (128)|29.6 (128)|33.54 (128)|37.74 (128)|
> |**LLaDA**|KLASS|**74.60** (82.50)|**32.2** (92.61)|**34.76** (79.04)|**40.08** (79.96)|
> |**Dream**|Top-k|72.18 (128)|33.6 (128)|42.68 (128)|47.47 (128)|
> |**Dream**|KLASS|**74.75** (118.92)|**37.6** (113.43)|**54.27** (68.03)|**63.04** (98.32)|
>
> *Each cell: Accuracy (Steps)*
>
> We will include above discussion and results in our final manuscript to strengthen the paper.
>
> ---
> ### **References**
> [1] Brock et al. "Large Scale GAN Training for High Fidelity Natural Image Synthesis." ICLR2019.
>
> [2] Zhou et al. "Activation Maximization Generative Adversarial Nets." ICLR2018.

---

> > ### Comment · Reviewer_yKia · 2025-08-05
> >
> > Thanks for the reply.
> > 1. The FID metric is important for generation task, reflecting distribution distance. The very marginal lift in FID limits the effectiveness of this method, especifically given that this paper adopts KL divergence for better performance. The lift in IS is also encouraged but not enough for verifying effectiveness.
> > 2. The model/task -specific hyperparameter search problem still exists. Though the general serach pipeline doesn't complicate this so much, this still significantly limiting the practicality. In contrast, the traditional confidence-guided strategy is more general and easy-to-use.

---

> ### Author Response · Authors · 2025-08-07
>
> ### 1. Effectiveness in MaskGIT
>
> We sincerely thank the reviewer for the constructive and insightful feedback. We acknowledge the reviewer's valid point regarding the importance of the FID metric and that the improvement presented initially might appear marginal. To better address this concern, we would like to offer a deeper analysis that we believe clarifies the core contribution and effectiveness of our proposed KLASS sampler. This analysis is directly motivated by a key finding within the original MaskGIT paper itself.
>
> In MaskGIT [1], the authors observed that for their confidence-based scheduling function, "more iterations are not necessarily better." They found that performance tends to peak at a sweet spot before worsening again. Crucially, they hypothesized that this is because "too many iterations may discourage the model from keeping less confident predictions, which worsens the token diversity." This points to a fundamental limitation of relying solely on confidence: it can lead to a loss of sample diversity (i.e., mode collapse) as the model prematurely converges on a narrow set of high-probability tokens, especially when more refinement steps are applied. By evaluating a prediction's stability rather than just its peak confidence, KLASS avoids the greedy, premature commitment that causes the baseline to fail. This preserves token diversity and enables stable, robust image generation even as the number of refinement steps increases.
>
> Our new results below demonstrate this point clearly. As the number of steps increases from 64 to 128, the baseline confidence scheduler's performance collapses, with its FID score worsening from 46.84 to 53.23. In stark contrast, KLASS not only delivers a significantly better FID at both settings, but its performance remains remarkably stable and robust.
>
>
> | Scheduler | Steps | FID (↓) |
> | :--- | :--- | :--- |
> | Confidence | 64 | 46.84 |
> | KLASS | 64 | **38.98** |
> | Confidence | 128 | 53.23 |
> | KLASS | 128 | **39.05** |
>
>
> Therefore, the primary contribution of KLASS lies in its ability to solve this fundamental stability-diversity issue, preventing the loss of sample diversity and performance collapse when using longer, more refined sampling schedules. This specific capability is what explains the superior and robust performance of KLASS on MaskGIT, as demonstrated in our experiments.
>
> To further substantiate that the effectiveness of KLASS is not confined to a narrow sweet spot where the original MaskGIT performs optimally, we will include expanded experimental results in the final manuscript, testing KLASS on additional generative models (e.g. MMaDA[2]).
>
>
>
> ### 2. Practicality of KLASS
>
> Thank you for raising this important concern regarding the practicality of our method.
> To address this concern, we present results using a single, non-optimized configuration of KLASS (confidence = 0.9, KL = 0.01) across all models and datasets. As shown below, this default setting consistently outperforms the top-confidence baseline in both accuracy and efficiency. This demonstrates that users can adopt KLASS with a robust default setting for immediate performance gains without any task-specific tuning, making it just as general and easy-to-use as traditional confidence-guided strategy, yet with significantly better performance.
>
>
> | Model | Method         | GSM8K              | MATH              | Humaneval         | MBPP               |
> | --- | --- | --- | --- | --- | --- |
> | LLaDA | Top-confidence | 72.40 (128)        | 29.6 (128)        | 33.54 (128)       | 37.74 (128)        |
> | LLaDA | KLASS          | **74.07** (118.29)        | **32.6** (117.43)        | **38.41** (86.34) | **38.91** (101.67) |
> | Dream | Top-confidence | 72.18 (128)        | 33.6 (128)        | 42.68 (128)       | 47.47 (128)        |
> | Dream | KLASS          | **73.31** (108.07) | **35.8** (110.13) | **54.27** (56.29) | **64.59** (94.37)  |
>
>
> For users seeking to maximize performance, KLASS accommodates an optional, lightweight tuning process. As detailed in our response [W1-1], this procedure requires only a small validation set (~100 examples) and is computationally inexpensive. For LLaDA on MATH500, for example, the tuning process takes around 4.2 hours, which is even less than the time required for a single inference run using a top-1 confidence baseline (5.3 hours). Therefore, KLASS offers a practical and generalizable sampling strategy that achieves strong performance without task-specific tuning, while also supporting lightweight tuning for additional gains.
>
>
> ---
> ### **References**
> [1] Chang et al. "MaskGIT: Masked Generative Image Transformer." arXiv:2202.04200 (2022).
>
> [2] Yang et al. "MMaDA: Multimodal Large Diffusion Language Models." arXiv:2505.15809 (2025).

---

> ### Author Response · Authors · 2025-08-09
>
> Dear Reviewer yKia,
>
> Thank you for the continued discussion, which has pushed us to clarify the effectiveness of our method. To directly address your valid concerns about the performance gains on MaskGIT [1], we have conducted a new experiment on MMaDA [2], a larger-scale (8B) and more recent state-of-the-art masked diffusion model that surpasses prior work like SDXL [3].
>
> For this experiment, we evaluated image generation performance by generating 10,000 images at 512x512 resolution from 1,000 ImageNet class prompts. For KLASS, we used a KL threshold of 0.1 and a confidence threshold of 0.3. The results, using MMaDA's default 15-step schedule for a fair comparison, are summarized below:
>
>
> | Scheduler | Steps | FID (↓) | IS (↑) |
> | :--- | :--- | :--- | :--- |
> | Confidence | 15 | 47.61 | 40.64 |
> | KLASS | 15 | **39.86** | **50.66** |
>
> As the results clearly demonstrate, KLASS provides a substantial improvement over the standard confidence-based sampler. Specifically, KLASS achieves a 7.75-point reduction in FID and a 10.02-point increase in IS. Achieving such a significant performance gain on a challenging, large-scale model like MMaDA is particularly meaningful, as it underscores the practical value and effectiveness of KLASS on current state-of-the-art systems. This result highlights the fundamental value of our stability-based criterion. We will conduct additional experiments using larger image sets and a wider range of step settings, and include the results in the manuscript.
>
> We believe this new evidence on a distinct and highly relevant model definitively demonstrates that the effectiveness of KLASS is not marginal, but substantial and generalizable. We hope this addresses your concerns and would be grateful to hear any further thoughts you might have.
>
> Thank you once again for your valuable feedback.
>
> ---
> ### **References**
> [1] Chang et al. "MaskGIT: Masked Generative Image Transformer." arXiv:2202.04200 (2022).
>
> [2] Yang et al. "MMaDA: Multimodal Large Diffusion Language Models." arXiv:2505.15809 (2025).
>
> [3] Podell et al. "SDXL: Improving Latent Diffusion Models for High-Resolution Image Synthesis." arXiv:2307.01952 (2023).

---

### Official Review · Reviewer_SCro · 2025-06-28

**Clarity:** 3
**Significance:** 2
**Originality:** 3
**Rating:** 5
**Confidence:** 3

**Summary:**

This paper presents KLASS, a method for accelerating sampling from masked diffusion models, based on estimating both the confidence and KL divergence between adjacent timesteps to identify stable tokens in which to unmask. Experimental results show consistent 2x speedups while maintaining performance.

**Questions:**

- What is "Halton-based" (line 247)?
- Is there a reason the appendix seems to just contain a completely different (assumed earlier) version of the paper?
- Lines 146-147: This should say "KL score for corrects tends to be significantly **lower**", right?

**Ethical Concerns:**

["NO or VERY MINOR ethics concerns only"]

**Final Justification:**

Overall, I am now confident in my score of accept. The paper's contribution is strong and straightforward, with clear reasoning and empirical results across the board. In the rebuttal, the author's addressed my only real concern with additional experiments and theoretical insight, and thus I think the paper is deserving of acceptance at NeurIPS.

**Limitations:**

See weaknesses.

**Quality:**

3

**Strengths And Weaknesses:**

**Strengths**
- Overall approach is both theoretically sound and empirically motivated, and is very intuitive to understand.
- Results of KLASS seem to flatly improve methods across modalities on all tested metrics, which clearly shows their approach works in practice.


**Weaknesses**
- Lines 144-146: I'm not sure if this is true? The plots seem to suggest that while correct samples have *slightly* higher confidence than incorrect samples across timesteps, they seem to be well within the variance bounds for significance (though proper error bars are not reported in lieu of a scatter plot, which I think could be changed). If anything, I think this argues that internal model confidence is *not* indicative of final correctness. Additionally, the error bounds for the KL divergence plot (1b) seem even larger, and it is hard to visually tell whether the plotted smoothing line is actually a reasonable fit. The proposed method obviously works, but more could be done to give explanation as to *why* empirically other than just this plot.
- I think the authors do not fairly recognize their limitation that while their speedup method is training-free, precisely because of that its overall acceleration ability is upper-bounded and cannot achieve the order-of-magnitude speedups that distillation methods get.

---

> ### Author Rebuttal · Authors · 2025-07-31
>
> We are deeply grateful for your thoughtful and constructive review, which has been invaluable in improving our work. We were particularly encouraged by your acknowledgment of:
>
> - The theoretical soundness, empirical motivation, and intuitive nature of our approach.
>
> - Its consistent performane improvements across all testeed modalites and metrics, demonstrating practical effectiveness.
>
> We have carefully considered the concerns you raised and provide our detailed responses below.
>
> ---
>
> ### [W2-1] Empirical Justification of the Stability Criterion
>
> We appreciate the reviewer’s insightful feedback regarding the interpretation of the plots and the relationship between internal model confidence, KL divergence, and correctness.
>
> To address this, we conducted a newly measured statistical analysis across four reasoning benchmarks, comparing average confidence and KL divergence between correct and incorrect samples for both LLaDA and Dream. As detailed in the tables below, correct samples consistently exhibit higher confidence and lower KL divergence compared to incorrect samples across all tasks and models. This consistent pattern across multiple benchmarks and both models provides empirical evidence that internal confidence and KL divergence serve as reliable indicators of correctness throughout the decoding process.
>
> **LLaDA: Confidence**
> | |GSM8K|MATH|HumanEval|MBPP|
> |---|---|---|---|---|
> |Correct|0.9626 ± 0.0104|0.9512 ± 0.0179|0.9756 ± 0.0136|0.9517 ± 0.0123|
> |Incorrect|0.9497 ± 0.0147|0.9253 ± 0.0263|0.9667 ± 0.0155|0.9450 ± 0.0201|
>
> **LLaDA: KL Divergence**
> | |GSM8K|MATH|HumanEval|MBPP|
> |---|---|---|---|---|
> |Correct|0.0620 ± 0.0161|0.0773 ± 0.0261|0.0449 ± 0.0226|0.0821 ± 0.0234|
> |Incorrect|0.0812 ± 0.0236|0.1091 ± 0.0376|0.0537 ± 0.0235|0.0928 ± 0.0319|
>
> **Dream: Confidence**
> | |GSM8K|MATH|HumanEval|MBPP|
> |---|---|---|---|---|
> |Correct|0.9417 ± 0.0176|0.9309 ± 0.0215|0.9813 ± 0.0138|0.9469 ± 0.0185|
> |Incorrect|0.9224 ± 0.0224|0.9218 ± 0.0248|0.9724 ± 0.0133|0.9204 ± 0.0335|
>
> **Dream: KL Divergence**
> | |GSM8K|MATH|HumanEval|MBPP|
> |---|---|---|---|---|
> |Correct|0.0743 ± 0.0215|0.0873 ± 0.0279|0.0254 ± 0.0154|0.0612 ± 0.0231|
> |Incorrect|0.0988 ± 0.0287|0.0933 ± 0.0319|0.0383 ± 0.0188|0.0891 ± 0.0394|
>
> We will include comprehensive results across all datasets in the final manuscript to further substantiate these findings.
>
> Furthermore, to provide a theoretical explanation for *why* these empirical patterns are reliable indicators of correctness, we have developed a new information-theoretic framework.
>
> We formalize the "unmasking risk" of a token $i$ using Conditional Mutual Information (CMI), $I_t(i) = I(X_i; Z \mid V_t)$, which measures the token's dependency on other unknown parts of the sequence. Our core theoretical result proves that the criteria used by KLASS (low cumulative KL divergence and high confidence) are sufficient to guarantee that this CMI is bounded by a provably value, $\Gamma_t(i) \le 0.6 \, \text{bits}$.
>
> In essence, this means a token's distributional stability—which we show empirically in the tables above—is a strong indicator of its informational independence. A token that is informationally independent is "safe" to unmask and highly likely to be correct, as its value does not depend on other unknowns. This provides the formal link between our empirical observations and the success of our method.
>
> For the full derivation and detailed discussion of this new framework, we kindly refer the reviewer to our response to Reviewer 5Uk3 [w4-1].
>
> ### [W2-2] Comparison with Distillation-Based Methods
> We agree that distillation can achieve greater speedups. However, we argue that our training-free method, KLASS, is a complementary and orthogonal approach, not a competing one.
>
> A distilled model still relies on an iterative sampling process, and KLASS, as a universal sampler, is designed to optimize this remaining step. This strategy of stacking optimizations has been shown to be effective; for example, Di4C [1] further accelerated the already-distilled SDTT [2] model.
>
> The two methods address different aspects of optimization. Distillation aims to create a new, faster model artifact through a costly, model-specific training procedure. In contrast, KLASS is a training-free sampler that optimizes the inference process for any given model. This makes KLASS a low-cost, universal, and plug-and-play tool that can be applied immediately for efficiency gains, complementing the more intensive approach of distillation.
>
> **[Q2-1]What is "Halton-based" (line 247)?**
>
> This refers to the experimental setup from the paper "Halton Scheduler for Masked Generative Image Transformer" [3] For our image generation task, we adapted the publicly released code for their confidence-based MaskGIT implementation to serve as our baseline.
>
> **[Q2-2]Is there a reason the appendix seems to just contain a completely different (assumed earlier) version of the paper?**
>
> Thank you for pointing it out. The slight differences you may have noted are intended to improve overall readability for the reviewers.
>
> **[Q2-3]Lines 146-147: This should say "KL score for corrects tends to be significantly lower", right?**
>
> Thank you for the careful reading and the helpful suggestion. You are correct. Our intended meaning is that the KL score for correct tokens is significantly lower, as shown in Figure 1b. We will revise the text to make this clearer in the final version.
>
> ---
> ### **References**
> [1] Hayakawa et al. "Distillation of Discrete Diffusion through Dimensional Correlations." ICML2025.
>
> [2] Deschenaux et al. "Beyond Autoregression: Fast LLMs via Self-Distillation Through Time." ICLR2025.
>
> [3] Besnier et al. "Halton Scheduler for Masked Generative Image Transformer" ICLR2025.

---

> > ### Comment · Reviewer_SCro · 2025-08-02
> >
> > I thank the authors for their response, which has generally addressed all my major concerns. I will maintain my score of accept.

---

> > > ### Author Response · Authors · 2025-08-07
> > >
> > > Dear reviewer SCro,
> > >
> > > We are glad that our response addressed your concerns. We sincerely appreciate your decision to recommend acceptance of our paper. Your careful review and insightful feedback were invaluable, and we will revise our paper to further clarify our work based on your comments. Thank you for your time and thoughtful engagement.

---

### Official Review · Reviewer_hkNH · 2025-07-03

**Clarity:** 3
**Significance:** 2
**Originality:** 2
**Rating:** 5
**Confidence:** 1

**Summary:**

This paper introduces KL-Adaptive Stability Sampling (KLASS), a novel training-free sampler designed to accelerate inference in masked diffusion models. The method dynamically unmasks multiple "stable" tokens in each iteration by leveraging token-level Kullback-Leibler (KL) divergence and model confidence. The authors provide theoretical justification and extensive empirical validation across various domains, including reasoning, text generation, image synthesis, and molecular generation, demonstrating significant speedups and, in many cases, improved performance.

**Questions:**

See weaknesses above.

**Ethical Concerns:**

["NO or VERY MINOR ethics concerns only"]

**Final Justification:**

updating score from 4 to 5 upon reading the rebuttal, but do note my low confidence in this topic.

**Limitations:**

yes

**Quality:**

2

**Strengths And Weaknesses:**

**Strengths**

* Novel and Intuitive Method: It intelligently exploits the internal dynamics of the diffusion process to identify which parts of the sequence have converged, allowing for more aggressive and efficient unmasking. This contrasts with more rigid, pre-defined scheduling or methods that require external "planner" models.

* Strong and Broad Empirical Validation: The paper's most significant strength lies in its comprehensive experimental evaluation. The authors test KLASS across a remarkably diverse set of tasks and models:
    1. Reasoning and Code Generation (GSM8K, MATH, HumanEval, MBPP): Demonstrates not just a 2x speedup but also an improvement in accuracy, which is a compelling result.
    2. Open-Ended Text Generation: Shows improved MAUVE scores and reduced perplexity, indicating higher-quality text generation.
    3. Image and Molecular Generation: The method's effectiveness is further solidified by its successful application to different modalities, achieving better FID/IS scores for images and higher reward metrics for molecules.

* Training-Free and Efficient: KLASS is a plug-and-play sampler that requires no additional training or architectural modifications. This is a significant practical advantage over methods that require auxiliary models, making it easy to adopt and implement with existing masked diffusion models.

* Theoretical Justification: The paper provides a theoretical argument to ground the heuristic. The theorem formally links suboptimal token choices to high KL divergence under context shifts, providing a principled reason why filtering by KL stability should lead to better and more robust generation.

* Thorough Ablation Studies: The authors include valuable ablation studies that analyze the interplay between confidence and KL thresholds and the benefit of multi-token unmasking over single-token strategies.

**Weaknesses**
* Hyperparameter Sensitivity: The effectiveness of KLASS appears to be dependent on several key hyperparameters: the KL threshold, the confidence threshold, and the history length. The paper provides the values used for each experiment but could benefit from a more in-depth analysis of the sensitivity to these parameters. A more detailed discussion on how to select these values for new tasks or models would enhance the practical utility of the paper.

* Computational Overhead of KL Divergence: While the paper argues that the overhead of computing KL divergence is minimal compared to the cost of a full diffusion step, this overhead is not zero. In scenarios with very large vocabularies, calculating KL divergence for every token at every step could become a non-trivial computational cost. A more detailed analysis of this overhead in practice, especially in relation to vocabulary size, would be beneficial.

* Comparison to Other Adaptive Sampling Methods: While the paper compares against several strong baselines, the field of accelerating diffusion models is rapidly evolving. The "Related Works" section could be strengthened by a more direct comparison to other contemporary adaptive sampling techniques, even if they are for continuous diffusion, to better contextualize the contribution. For instance, methods that dynamically adjust the number of sampling steps based on other criteria could be relevant points of comparison.

* Were the inference time costs measured in the experiments? It would be insightful to report wall clock times.

Typo on line 142 ("reasoining")

---

> ### Author Rebuttal · Authors · 2025-07-31
>
> We sincerely appreciate your insightful review and are grateful for your acknowledgement that our work:
> - Proposes a novel and intuitve method
> - Provides strong and broad empricial validation across diverse domains, demonstrating both significant speedups and peformance improvements.
> - Offeres a practical, training-free and plug-and-play sampler.
> We have carefully considered your concerns and provide our detailed responses below.
> ---
>
> ### [W1-1] Hyperparameter Sensitivity
> We clarify below how we select hyperparameters in practice and provide additional evidence that KLASS is robust to variations around the chosen values.
>
> **1. Hyperparameter Selection Guideline**
>
> For hyperparameter selection process, we follow a lightweight three-step search procedure using a small validation set, around 100 examples.
>
> - **Step 1. Initial KL Threshold Estimation**:
> We estimate a reasonable KL threshold by inspecting the distribution of KL values during decoding.
> - **Step 2. Confidence Threshold Search**:
> With a fixed KL threshold, we sweep confidence values (e.g., 0.9 down to 0.6) to find the best balance of accuracy and decoding speed.
> - **Step 3. KL Threshold Refinement**:
> With a fixed confidence threshold, we perform a finer search for the optimal KL value around the initial estimate.
>
> This process is fast, requiring only a small number of samples and negligible computation compared to training.
>
> **2. Hyperparameter Sensitivity Grid Search**
>
> To address sensitivity, we performed a grid search across diverse models and tasks (LLaDA/Dream for reasoning, MDLM for molecular), demonstrating KLASS's robustness.
>
> Our findings show that configurations near the selected optimum consistently yield high accuracy with significantly reduced sampling steps. For instance, on HumanEval with LLaDA, settings near (confidence=0.9, KL=0.01) maintain or improve upon the baseline accuracy of **39.63%** while using far fewer than its 256 steps. Moreover, these searches often reveal that slightly different settings can outperform our main reported configuration—such as (confidence=0.95, KL=0.005) on MBPP with Dream—indicating both robustness and potential for further tuning.
>
> **HumanEval (LLaDA)** : Conf = 0.9, KL = 0.01
> \ **Baseline Performance**: 39.63 (NFE=256)
> |**Conf**|KL = 0.015|**KL = 0.01**|KL = 0.005|
> |---|---|---|---|
> |0.95|39.63 (65.19)|40.24 (99.29)|40.24 (103.14)|
> |**0.9**|40.24 (89.29)|**40.85 (91.98)**|40.24 (96.75)|
> |0.85|39.63 (84.98)|39.63 (88.78)|40.24 (94.48)|
> |0.8|39.02 (83.14)|39.02 (87.21)|40.85 (94.01)|
>
> **MBPP (LLaDA)**: Conf = 0.7, KL = 0.01
> \ **Baseline Performance**: 48.64 (NFE=256)
> |**Conf**|KL = 0.015|**KL = 0.01**|KL = 0.005|
> |---|---|---|---|
> |0.8|49.42 (122.37)|49.42 (134.52)|49.42 (134.52)|
> |0.75|49.03 (118.61)|49.42 (123.45)|48.25 (132.11)|
> |**0.7**|48.25 (116.24)|**49.03 (127.81)**|49.03 (130.67)|
> |0.65|46.30 (113.22)|48.64 (118.54)|49.42 (128.99)|
>
> **HumanEval (Dream)**: Conf = 0.8, KL = 0.001
> \ **Baseline Performance**: 58.53 (NFE=256)
> |**Conf**|KL = 0.005|KL = 0.003|**KL = 0.001**|KL = 0.0005|
> |---|---|---|---|---|
> |0.9|54.87 (64.78)|56.10 (67.41)|57.93 (74.86)|61.59 (79.57)|
> |0.85|57.32 (59.39)|55.49 (65.31)|59.15 (73.38)|60.98 (79.43)|
> |**0.8**|51.22 (58.09)|54.27 (62.82)|**59.76 (73.73)**|60.96 (79.51)|
> |0.75|48.78 (55.21)|53.05 (62.61)|59.15 (73.41)|60.37 (79.37)|
>
> **MBPP (Dream)**: Conf = 0.9, KL = 0.001
> \ **Baseline Performance**: 63.81 (NFE=256)
> |**Conf**|KL = 0.005|KL = 0.003|**KL = 0.001**|KL = 0.0005|
> |---|---|---|---|---|
> |0.95|65.37 (108.93)|65.37 (110.03)|64.59 (112.56)|64.59 (113.14)|
> |**0.9**|62.65 (103.99)|64.20 (107.09)|**64.59 (111.24)**|64.59 (112.65)|
> |0.85|64.20 (101.34)|63.81 (105.43)|65.37 (112.54)|64.59 (112.76)|
> |0.8|63.81 (96.02)|63.04 (103.22)|65.37 (112.52)|64.59 (112.82)|
>
> *Each cell: Accuracy (Steps)*
>
> **Molecule (QED)**: Conf = 0.98, KL = 0.001
> \ **Baseline Performance**: 0.526 (NFE=32)
> |**Conf**|KL = 0.01|KL = 0.005|**KL = 0.001**|KL = 0.0005|
> |---|---|---|---|---|
> |0.999|0.527 (18.607)|0.526 (18.611)|0.524 (18.580)| 0.538 (18.452)|
> |0.99|0.515 (18.223)|0.521 (18.325)|0.543 (18.689)| 0.537 (18.664)|
> |**0.98**|0.531 (18.434)|0.517 (18.377)|**0.546** (18.782)| 0.534 (18.823)|
> |0.96|0.529 (18.431)|0.535 (18.511)| 0.534 (18.634)| 0.543 (18.801)|
>
> *Each cell: QED (Steps)*
>
> We will include the results for all datasets in our final version of the manuscript.
>
> ### [W1-2] Computational Overhead of KL Divergence
> The overhead is minimal because it is a lightweight post-processing step on existing logits, requiring no additional forward pass. For the set of masked tokens $I_m = \lbrace i \mid z^i_t = m \rbrace$, the process computes the KL score $d^i_t = D_{KL}(p^i_t || p^i_{t-1})$ and caches the prior distribution. This yields a combined computational $\Delta C$ and memory overhead of $O(|I_m| \cdot |V|)$. This linear cost is negligible, as the simple vector operations are insignificant compared to the expensive matrix multiplications and multi-gigabyte footprint of the main diffusion step.
>
> Empirically, our wall-clock time measurements validate this conclusion, showing **memory overheads under 1.57%** and **latency overheads under 0.21%** per decoding step. Additional results for sequence lengths up to 1,024 are provided in [W4-4].
>
> |Model|Vocab Size|Length|Memory Overhead (MB)|Total Memory (MB)|Time Overhead (s)|Total Time per Step (s)|
> |---|---|---|---|---|---|---|
> |**LLaDA**|126,464|256|247|18,702|0.000255|0.1218|
> |**Dream**|152,064|256|296|18,875|0.000177|0.1275|
>
> We will include this analysis in our revised manuscript.
>
> ### [W1-3] Comparison to Other Adaptive Sampling Methods
> We thank the reviewer for the valuable suggestion. We agree that a broader discussion of adaptive sampling methods, including those for continuous diffusion, will better contextualize our work. We will incorporate a full comparative analysis, summarized below, into our final manuscript.
>
> KLASS operates dynamically, creating a sample-specific schedule on-the-fly guided by an internal measure of temporal stability (low KL divergence). This self-planning approach contrasts with methods that use fixed, pre-computed schedules from theoretical error bounds (e.g., JYS [1], DUS [2]) or rely on heavy external models for path guidance and error-correction remasking (P2 [3], DDPD [4]). Furthermore, its stability criterion advances beyond simpler heuristics like low confidence (ReMDM [5]) or high entropy (Diffusion-EAGS [6]) used for single-token correction, as it allows KLASS to finalize multiple converged tokens at once to efficiently shorten the generation path.
>
> Furthermore, this step-reduction strategy is orthogonal and complementary to techniques that reduce per-step computational cost. Methods like caching (FreeCache [7], dKV-Cache [8]), which make each denoising step cheaper, can be combined with KLASS to compound acceleration.
>
> While KLASS shares an adaptive spirit with solvers for continuous diffusion, their mechanisms are fundamentally different. Continuous solvers like DDIM [9] and DPM-Solver [10] reframe the reverse process as an ODE, using numerical methods to modulate a global sampling trajectory's step size based on local error, enabling 10-20 step generation. In contrast, KLASS operates in the discrete domain by adapting the inference lifespan of individual tokens—a mechanism tailored for token-level convergence.
>
> In summary, KLASS is a lightweight, training-free sampler that uses a unique temporal stability criterion to dynamically shorten inference, setting it apart from methods that rely on fixed schedules, external planners, simpler heuristics, or orthogonal computational optimizations.
>
> ### [W1-4] Wall Clock Time
> Yes, we have reported wall-clock latency for MATH in the appendix (Table 8). We have now newly measured and included results for all datasets:
>
> |**Model**|**Dataset**|**Top-1 Acc**|**KLASS Acc**|**Top-1 Time (s)**|**KLASS Time (s)**|**Speedup**|
> |---|---|---|---|---|---|---|
> |**LLaDA**|GSM8K|75.13|76.50|37.04|15.86|**2.34x**|
> |**LLaDA**|MATH|31.40|33.80|38.40|21.41|**1.79x**|
> |**LLaDA**|Humaneval|39.63|40.85|39.54|16.04|**2.47x**|
> |**LLaDA**|MBPP|46.69|47.86|39.12|20.68|**1.89x**|
> |**Dream**|GSM8K|79.75|80.44|29.66|22.26|**1.33x**|
> |**Dream**|MATH|38.00|43.20|30.76|23.31|**1.32x**|
> |**Dream**|Humaneval|58.53|59.76|32.01|11.52|**2.78x**|
> |**Dream**|MBPP|63.81|64.59|31.89|17.65|**1.81x**|
>
> These results validate that KLASS achieves substantial speedups with comparable or improved accuracy across diverse reasoning tasks. Notably, we observe over 2× speedup on GSM8K for LLaDA, and over 2× speedup on HumanEval for both models, confirming that our method provides meaningful efficiency gains in practical settings.
>
> **[Typo]** Thank you for pointing it out. We will thoroughly proofread and refine the final version.
>
> ---
> ### **References**
> [1] Park et al. "Jump Your Steps: Optimizing Sampling Schedule of Discrete Diffusion Models." ICLR2025.
>
> [2] Luxembourg et al. "Plan for Speed: Dilated Scheduling for Masked Diffusion Language Models." arXiv:2506.19037 (2025).
>
> [3] Peng et al. "Path Planning for Masked Diffusion Model Sampling." arXiv:2502.03540 (2025).
>
> [4] Liu et al. "THINK WHILE YOU GENERATE: DISCRETE DIFFUSION WITH PLANNED DENOISING." ICLR2025.
>
> [5] Wang et al. "Remasking Discrete Diffusion Models with Inference-Time Scaling." arXiv:2503.00307 (2025).
>
> [6] Koh et al. "PLM-Based Discrete Diffusion Language Models with Entropy-Adaptive Gibbs Sampling." arXiv:2411.06438 (2024).
>
> [7] Hu et al. "Accelerating Diffusion Language Model Inference via Efficient KV Caching and Guided Diffusion." arXiv:2505.21467 (2025).
>
> [8] Ma et al. "dKV-Cache: The Cache for Diffusion Language Models." arXiv:2505.15781 (2025).
>
> [9]Song et al. "Denoising Diffusion Implicit Models." ICLR2021.
>
> [10]Lu et al. "DPM-Solver: A Fast ODE Solver for Diffusion Probabilistic Model Sampling in Around 10 Steps." NeurIPS2022.

---

### Author Response · Authors · 2025-08-09
**General Response**

Dear reviewers and AC,

We sincerely thank all the reviewers for the valuable feedback and positive comments, which have greatly helped us improve and strengthen our work.

Our paper introduces KL-Adaptive Stability Sampling (KLASS), a method to accelerate inference in masked diffusion models. As highlighted by the reviewers, our key contributions are:

- **Novel and Intuitive**: Exploits the internal dynamics by leveraging token stability via low KL divergence to detect convergence. (hkNH, SCro, 5Uk3)
- **Practical and Training-Free**: A plug-and-play sampler requiring no training, architectural changes, or external planners. (hkNH, 5Uk3)
- **Broad Empirical Gains**: Improves or maintains quality across diverse tasks, with significant speedups. (hkNH, SCro, yKia, 5Uk3)
- **Theoretically Grounded**: Supported by an theoretic explanation linking high KL divergence to suboptimal token choices. (hkNH, SCro, yKia)

In response to reviewers’ constructive questions and suggestions, we have added the following:

- **Demonstrating Practicality:** We provided a lightweight hyperparameter search guideline with extensive sensitivity analyses. (hkNH [W1-1], yKia [W3-3], 5Uk3 [W4-4]) Also showed that KLASS outperforms the baseline even with non-optimized configurations without tuning. (yKia [W3-3])
- **Real-World Speed Evaluations:** Reported comprehensive wall-clock measurements showing up to 2.78× speedups. (hkNH [W1-4], 5Uk3 [W4-2])
- **Computational Overhead Analysis:** Confirmed negligible runtime and memory cost. (hkNH [W1-2], 5Uk3 [W4-5])
- **Empirical Justification of KLASS:** Added statistical analyses showing correct tokens consistently exhibit higher confidence and lower KL divergence. (SCro [W2-1])
- **Theoretical Improvement:** Developed an information-theoretic proof that formally guarantees the safety of KLASS’s parallel unmasking. (5Uk3 [W4-1], SCro [W2-1])
- **Effectiveness on Image Generation:** Validated KLASS's effectiveness with two key results: preventing performance collapse on MaskGIT with longer schedules, and achieving substantial FID gains on the canonical MMaDA model. (yKia [W3-2])

We are grateful that most of our clarifications and additional experiments successfully addressed reviewers concerns, leading to positive outcomes including notably strong support. All these modifications and additional analyses will be reflected in our final manuscript accordingly. We hope that KLASS will inspire further research on sampling, contributing to the development of more efficient and reliable masked diffusion models.

Authors

---

### Note · Authors · 2025-08-13

Dear AC, SAC and reviewers,

We are grateful that the reviewers found our method a “delightfully simple yet powerful remedy” and “conceptually elegant,” noting that it “extracts extra mileage from the model’s own logits” (5Uk3) and “intelligently exploits the internal dynamics of the diffusion process” (hkNH). Reviewers further described it as “theoretically sound,” “intuitive” (SCro), and “rational” (yKia), highlighted its “significant practical advantage” (hkNH), and judged it “likely to be adopted quickly by practitioners” (5Uk3). Reviewers also praised the “comprehensive experimental evaluation” (hkNH), “effectively verified” results (yKia), and “unusually broad” empirical study (5Uk3).

We thank the reviewers for raising important concerns that helped strengthen the work. Below we summarize the concerns and our responses:

**Reviewer hkNH**. Asked about (1) hyperparameter sensitivity, (2) KL-divergence overhead, (3) comparisons with other adaptive sampling methods, and (4) wall‑clock times. We showed low sensitivity with a lightweight search guideline, negligible time and memory overhead for computing token‑level KL, clarified methodological differences, and reported wall‑clock times.

**Reviewer SCro**. Requested (1) stronger empirical justification and (2) comparison to distillation‑based methods. We added statistical analyses of confidence and KL with theoretical grounding, and clarified that KLASS is complementary and orthogonal to distillation.

**Reviewer yKia**. Asked about (1) the KL–confidence relationship, (2) the marginal lift in MaskGIT, and (3) practicality. We showed low correlation between KL and confidence, larger gains at longer schedules for MaskGIT, additional improvements with the advanced MMaDA model, and that KLASS outperforms the baseline without tuning; optional tuning yields further gains at low cost.

**Reviewer 5Uk3**. Requested (1) stronger theoretical guarantees, (2) wall‑clock latency, (3) variance estimates, (4) hyperparameter sensitivity, and (5) KL‑history overhead. We strengthened the theory and addressed all other points, which led to a *strong* accept recommendation.

We believe our comprehensive new evidence successfully resolves all raised concerns and substantively strengthens our work. We appreciate the careful consideration of all reviewers, ACs, and SACs.


Sincerely, \
Authors

---

### Decision · Program_Chairs · 2025-09-17

**Decision:**

Accept (spotlight)

**Comment:**

This paper introduces KL-Adaptive Stability Sampling (KLASS), a simple yet powerful method to accelerate inference in masked diffusion models by leveraging token-level KL divergence and confidence to identify stable tokens for early unmasking. Reviewers praised the method as conceptually elegant, intuitive, and practically impactful, noting its broad empirical validation across reasoning, text, image, and molecule generation tasks. Concerns about hyperparameter sensitivity, computational overhead, and comparisons with related work were thoroughly addressed in the rebuttal with new analyses, statistical validation, and expanded experiments, including strong results on large-scale models. The value is further enhanced by the training-free, plug-and-play nature of KLASS, its solid theoretical grounding, and its consistent speedups of up to ~2–3× with maintained or improved quality. With multiple reviewers raising their scores after rebuttal (including one to a strong accept), the consensus is highly positive.

Overall, KLASS is a rare example of a simple, training-free method that yields consistent 2× speedups across diverse masked diffusion tasks, with no quality loss and negligible overhead. Its practicality and broad applicability make it highly likely to see real-world adoption and inspire follow-up work. I recommend it as a spotlight paper.